# Neural Concept Binder

**Wolfgang Stammer**[1,2,*]     **Antonia Wüst**[1,*]     **David Steinmann**[1,2,*]
**Kristian Kersting**[1,2,3,4]

[1]Computer Science Department, TU Darmstadt; [2]Hessian Center for AI (hessian.AI);
[3]German Research Center for AI (DFKI); [4]Centre for Cognitive Science, TU Darmstadt

## Abstract

The challenge in object-based visual reasoning lies in generating concept representations that are both descriptive and distinct. Achieving this in an unsupervised manner requires human users to understand the model's learned concepts and, if necessary, revise incorrect ones. To address this challenge, we introduce the *Neural Concept Binder* (NCB), a novel framework for deriving both discrete and continuous concept representations, which we refer to as "concept-slot encodings". NCB employs two types of binding: "soft binding", which leverages the recent SysBinder mechanism to obtain object-factor encodings, and subsequent "hard binding", achieved through hierarchical clustering and retrieval-based inference. This enables obtaining expressive, discrete representations from unlabeled images. Moreover, the structured nature of NCB's concept representations allows for intuitive inspection and the straightforward integration of external knowledge, such as human input or insights from other AI models like GPT-4. Additionally, we demonstrate that incorporating the hard binding mechanism preserves model performance while enabling seamless integration into both neural and symbolic modules for complex reasoning tasks. We validate the effectiveness of NCB through evaluations on our newly introduced CLEVR-Sudoku dataset. Code and data at: project page.

## 1 Introduction

An essential aspect of visual reasoning is obtaining a proper *conceptual* understanding of the world by learning visual concepts and processing these into a suitable representation (*cf.* Fig. 1). The majority of current machine learning (ML) approaches that focus on visual concept-based processing utilize forms of supervised [34, 68, 32, 84], weakly-supervised [41, 69, 49, 63, 8, 82] or text-guided [29] learning of concepts. These approaches all require some form of additional (prior) knowledge about the relevant domain. An attractive alternative, though much more challenging, is to learn concepts in an unsupervised fashion. This comes with several challenges: (i) learning an expressive concept representation without concept supervision is intrinsically difficult [40], and (ii) there is no guarantee that learned concepts align with general domain knowledge [36, 85, 9] and (iii) can therefore be utilized for complex downstream tasks. Moreover, (iv) to trust that the learned concept representations are reliable for high stakes scenarios [15], it is necessary to make the model's concept representations human-*inspectable* and -*revisable* [68, 69, 31] (*cf.* Fig. 1 (left)).

These challenges raise questions about the nature of the unsupervised learned concept representations. Continuous encodings [60, 59, 77, 79] are easier to learn and more expressive. However, they are difficult to interpret and suffer from problems related to poor generalization [81] and information leakage [47, 50]. On the other hand, discrete encodings [69, 79, 26, 4] are hard to learn [44, 72, 9], but are easier to understand and thus align, *e.g.*, to a task at hand.

---

*These authors share equal contribution.
Correspondence to: Wolfgang Stammer <wolfgang.stammer@cs.tu-darmstadt.de>.

38th Conference on Neural Information Processing Systems (NeurIPS 2024).

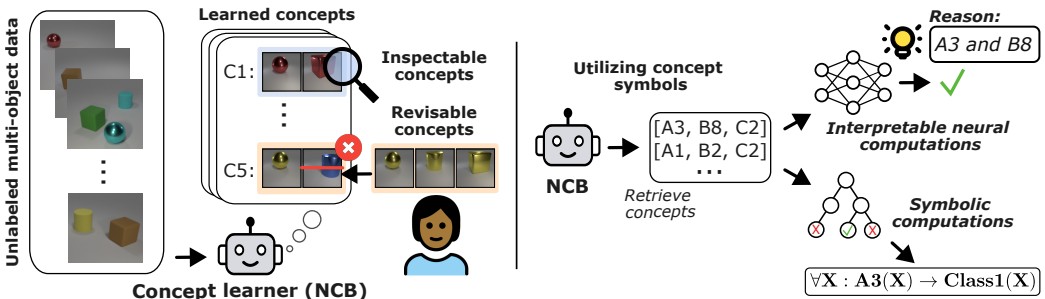

Figure 1: **Unsupervised learning of concepts for visual reasoning.** (left) Models that learn concepts from unlabeled data require inspectable and revisable concept representations. (right) Concepts obtained from the Neural Concept Binder (NCB) can be utilized both in (interpretable) neural and symbolic computations.

This work proposes the **Neural Concept Binder** (NCB) framework to learn expressive, yet inspectable and revisable, concepts from unlabeled data. NCB combines continuous encodings, obtained via block-slot-based *soft-binding*, with discrete concept representations, derived through retrieval-based *hard-binding*. NCB's soft binding leverages the object-factor disentanglement capabilities of the recent SysBinder mechanism [65]. Subsequently, NCB's hard binding mechanism utilizes HDB-SCAN [12, 13] to cluster the continuous block-slot encodings, distilling a structured corpus of discrete concepts from these clusters. This corpus enables the retrieval of discrete concept representations during inference by matching the continuous encoding with the closest entries in the corpus. Thus, to address the challenges of unsupervised concept learning, NCB integrates the strengths of both continuous *and* discrete concept representations. Moreover, NCB enables straightforward concept inspection and facilitates easy revision procedures, allowing alignment of the learned concepts with prior knowledge. In our evaluations, we demonstrate that NCB's discrete *concept-slot* encodings retain the expressiveness of their continuous counterparts. Moreover, they can be seamlessly integrated into downstream applications via symbolic *and* interpretable neural computations (*cf.* Fig. 1 (right)). In this context, we introduce our novel *CLEVR-Sudoku* dataset, which presents a challenging visual puzzle that requires both perception and reasoning capabilities (*cf.* Fig. 4).

In summary, our contributions are the following: **(i)** we introduce the Neural Concept Binder framework (NCB) for unsupervised concept learning, **(ii)** we show the possibilities to integrate NCB with symbolic and subsymbolic modules in challenging downstream tasks, achieving performance on par with supervised trained models, **(iii)** we highlight the possibilities of easy concept inspection and revision via NCB, and **(iv)** we introduce the novel CLEVR-Sudoku dataset, which combines challenging visual perception and symbolic reasoning.

## 2    Related Work

**Unsupervised visual concept learning** focuses on obtaining concept-level representations from unlabeled images [25]. Some works have tackled this only for specific domains, such as extracting "teachable" concepts for chess [62] or learning manipulation concepts from videos of task demonstrations [39]. Others rely on object-level concept guidance through initial image segmentations [27] or "natural supervision" [49]. In contrast, Vedantam et al. [77] and Wüst et al. [81] focus on learning higher-level relational concepts, *i.e.*, assuming that basic-level concepts have already been provided. Several approaches learn concepts from the training signal of an image classification task [78, 1, 14, 38], often focusing on image-region-based concepts [22]. More recently, several works have explored leveraging the knowledge stored in large pretrained models, such as combining large language models with CLIP embeddings [82, 52] or using weakly-supervised queries to a vision-language model [8]. These approaches still rely on some form of supervision, whether through text, class labels, or prompts. In contrast, this work focuses on learning unsupervised concepts at both the object and factor levels, ensuring that these concepts remain inherently inspectable and revisable.

The motivation for **inherently inspectable and revisable concept representations** is to allow human stakeholders to investigate and potentially revise a model's internal concepts. Most research in this area focuses on post-hoc approaches that distill concept knowledge from pretrained models [83, 21, 57,

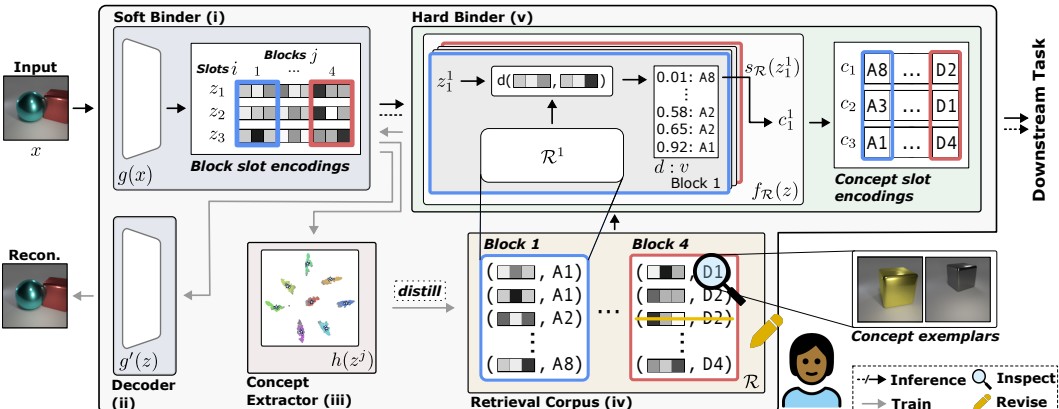

Figure 2: The **Neural Concept Binder** (NCB) combines continuous, block-slot encodings via slot-attention based image processing with discrete, concept-slot encodings via retrieval-based inference. The structured retrieval corpus (distilled from the block-slot encodings) allows for easy concept inspection and revision by human stakeholders. Moreover, the resulting concept-slot encodings can be easily integrated into complex downstream tasks.

18, 23]. In contrast, Lage and Doshi-Velez [35] explore learning inspectable concept representations through human feedback, focusing on tabular data and higher-level concepts. Similarly, Stammer et al. [69] develop inherently inspectable visual concepts using weak supervision and a prototype-based binding mechanism. However, no existing work addresses the development of inherently inspectable and revisable concept representations in the context of unsupervised visual learning.

The properties of **discrete vs. continuous encodings** are a vibrant research topic that is highly relevant to learning suitable concept representations. Continuous encodings allow for easier and more flexible optimization and information binding [43, 64, 65, 7]. However, discrete representations are considered essential for understanding AI models [31], mitigating shortcut learning [68, 3], and solving complex visual reasoning tasks [26, 66]. Despite their advantages, learning discrete representations through neural modules remains a challenging problem [44, 24, 20, 74]. While some works have focused on categorical-distribution-based discretization [4, 28, 46], others have explored retrieval-based discretization of continuous encodings using various forms of inherent "codebooks" [73, 71]. Only a few studies have explicitly addressed how to bind semantic visual information to specific discrete representations [69]. Whereas previous works typically emphasize one of the two representation types, we see great potential in the recent trend of explicitly integrating both discrete and continuous representations [17, 32, 84, 51].

## 3   Neural Concept Binder (NCB): Extracting Hard from Soft Concepts

In this work, we refer to a concept as "the label of a set of things that have something in common" [2]. This definition can be applied on different scales of a visual scene: on an image level (*e.g.*, an image of a *park*), an object level (*e.g.*, a *tree* vs. a *bird*) or an object-factor level (*e.g.*, the *color* of a bird). Our proposed Neural Concept Binder (NCB) framework tackles the challenge of learning inspectable and revisable object-factor level concepts from unlabeled images by combining two key elements: (i) continuous representations via SysBinder's block-slot-attention [65, 43] with (ii) discrete representations via retrieval-based inference. Fig. 2 provides an overview of NCB's inference, training, concept inspection, and revision processes. Let us formally introduce these processes.

Overall, we consider a set of *unlabeled* images $X := (x_1, \cdots, x_N) \in \mathbb{R}^{N \times D}$ with $x_i \in \mathbb{R}^D, N \in \mathbb{N}$ and $D \in \mathbb{N}$ (for simplicity, we drop the image index notation in the following). Briefly, given an image, x, NCB infers latent block-slot encodings, z, and performs a retrieval-based discretization step on z to infer concept-slot encodings, c. These express the concepts of the objects in the image, i.e., object-factor level concepts. We begin by introducing the inference procedure of NCB. We hereby assume that NCB's components have already been trained and will introduce details of the training procedure subsequently.

## 3.1 Inferring Concept-Slot Representations

**Obtaining *Continuous* Block-Slot-Encodings.** Consider an image $x \in X$. The first component of NCB, the *soft binder*, is based on the systematic binding mechanism [65] and is represented by a block-slot encoder (*cf.* Fig. 2 (i)), $g_\theta : x \to z \in \mathbb{R}^{N_S \times N_B \times D_B}$, where $g$ is parameterized by $\theta$ (for simplicity, this notation is omitted in the following). The soft binder transforms an input image into a latent, continuous *block-slot* representation, where $N_S$ represents the number of slots, $N_B$ the number of blocks per slot, and $D_B$ the dimension of each block. The soft binder employs two key types of *binding* mechanisms: spatial and factor binding. Spatial binding ensures spatial modularity across the entire scene and is achieved through slot attention [43], allowing each object in the image to be represented in a specific slot, $z_i$. Factor binding, introduced by Singh et al. [65], ensures that different object *factors* (*e.g.*, attributes like color) are encoded in separate blocks of a slot, *i.e.*, $z_i^j$. These two binding mechanisms work together to perform object- and factor-based image processing. We refer to Suppl. A.1 for additional details on both systematic (factor) binding and slot attention. Overall, the resulting block-slot encodings represent continuous, object-centric representations of the input image, with objects encoded in slots and object factors encoded within the blocks of those slots.

**Obtaining *Discrete* Concept-Slot-Encodings.** The role of NCB's second processing component, the *hard binder*, is to transform the continuous block-slot encodings into expressive, yet *discrete* concept-slot encodings. Specifically, the hard binder is represented by a retrieval encoder, $f$ (*cf.* Fig. 2 (v)), which processes the block-slot encodings, $z$, into a set of discrete concept-slot encodings, $c$. In detail, $f$ defines a function $f_\mathcal{R} : z \to c \in \mathbb{N}^{N_S \times N_B}$, parameterized by a retrieval corpus $\mathcal{R}$ (*cf.* Fig. 2 (iv)). This retrieval corpus consists of a tuple of sets $\mathcal{R} := [\mathcal{R}^1, \ldots, \mathcal{R}^{N_B}]$, where each set $\mathcal{R}^j := \{(\text{enc}_l^j, v_l) : l \in \{1, ..., |\mathcal{R}^j|\}\}$ contains tuples of block encodings, $\text{enc}_l^j \in \mathbb{R}^{D_B}$, and corresponding discrete values, $v_l \in \{1, \cdots, N_C\}$. Importantly, $\text{enc}_l^j$ is a representative block encoding of a specific *concept* cluster, determined during NCB's training phase (*cf.* Fig. 2 (iii), detailed below). $v_l$ serves as the *symbol* identifier for the concept cluster associated with $\text{enc}_l^j$. Each block can contain up to $N_C \in \mathbb{N}$ different concepts. To infer the concept symbol for a sample's block-slot encoding, NCB compares $z_i^j$ with the encodings in the corresponding block's retrieval corpus, $\mathcal{R}^j$, and selects the most fitting concept. Specifically, given a distance metric $d(\cdot, \cdot)$ and the block-slot encoding $z_i^j$, the selection function $s_\mathcal{R} : z_i^j \to l \in \mathbb{N}$ (Fig. 2 (v)) finds the index $l$ of the closest encoding in the retrieval corpus: $s_\mathcal{R}(z_i^j) = \text{argmin}_l \, d(\text{enc}_l^j, z_i^j)$ such that $(\text{enc}_l^j, v_l) \in \mathcal{R}^j$. This results in the concept representation for slot $i$ and block $j$, denoted as $c_i^j := v_{s_\mathcal{R}(z_i^j)}$. For slot $i$, the full concept representation is denoted as $c_i := [c_i^1, \ldots, c_i^{N_B}]$ and the final concept-slot encoding as $c := f_\mathcal{R}(z) = [c_1, \ldots, c_{N_S}]$. We refer to Suppl. A.2 for details on an alternative *top-k* selection function. We further note that NCB's flexibility, in principle, allows also to utilize the continuous encodings of its soft binder (Fig. 2 dashed arrow) in case a downstream task requires it. Let us now move on to NCB's training procedure.

## 3.2 Unsupervised Concept Learning via NCB

The training procedure of the Neural Concept Binder is separated into two subsequent steps where we provide an overview here and details in Suppl. A.3. We formally describe these steps using the pseudo-code in Alg. 1. The first step consists of optimizing the encoder, $g$, to provide *object-factorised* block-slot encodings. It is optimized for unsupervised image reconstruction based on the decoder model, $g'_{\theta'} : z \to \tilde{x} \in \mathbb{R}^D$ (Fig. 2 (ii)) and utilizing a mean squared error loss: $L = L_{\text{MSE}}(x, g'(g(x)))$. The goal of NCB's second training step is to obtain the retrieval corpus, $\mathcal{R}$. This procedure is based on obtaining an optimal clustering of block encodings via an unsupervised clustering model, $h$, and distilling the resulting information from $h$ into explicit representations in the retrieval corpus. For each block $j$ a clustering model $h_{\phi^j}$ (Fig. 2 (iii)) is fit to identify a potentially overparameterised set of clusters within a set of block encodings (based on an unsupervised criterion, *e.g.*, a density-based score [53]), resulting in $N_C \in \mathbb{N}$ clusters. Next, for each cluster, $v \in \{1, \cdots, N_C\}$, representative block encodings, $\text{enc}^j$, are extracted from $h$. Such an encoding represents either an averaged *prototype* or instance-based *exemplar* encoding. The corresponding tuples $(\text{enc}^j, v)$ are explicitly stored in the retrieval corpus $\mathcal{R}^j$ (Fig. 2 (iv)) where we use the index $l$ to identify specific encodings in $\mathcal{R}^j$, leading to $\mathcal{R}^j := \{(\text{enc}_l^j, v_l) : l \in \{1, ..., |\mathcal{R}^j|\}\}$. Thus, $\text{enc}_l^j$ represents one block encoding of $\mathcal{R}^j$ that has been assigned to cluster $v_l$. Finally, $\mathcal{R} = [\mathcal{R}^1, \cdots, \mathcal{R}^{N_B}]$ represents the final retrieval

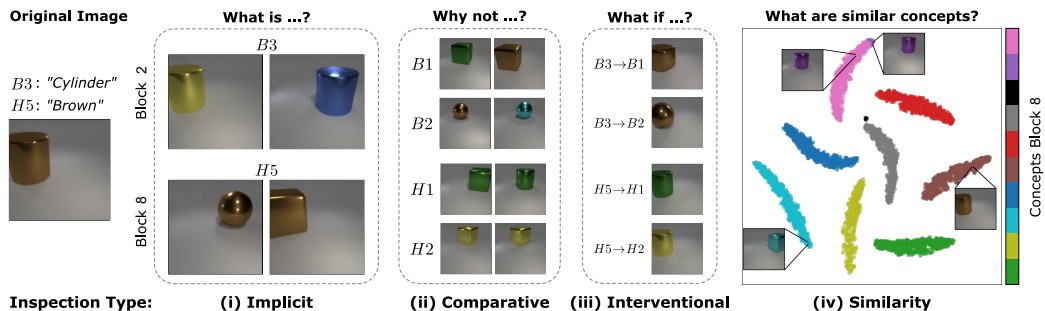

Figure 3: **NCB's concept space is inherently inspectable.** A human stakeholder can easily inspect the concept space by asking a diverse set of questions. For example, NCB answers interventional questions (iii) via generating images with selectively modified concepts.

corpus, *i.e.*, the set of corpora for each block. Through this training procedure, NCB learns to unsupervisedly categorize the object-factor information from the latent encoding space of the soft binder and stores this information in a structured, symbolic, and accessible way in the hard binder's retrieval corpus. We refer to the resulting clusters of each block as NCB's *concepts* and denote concepts with a capital letter for the block and a natural number for the category id, *e.g.*, $A3$. We note that in practice, it is further possible to finetune the block-slot encoder, $g$, through supervision from the hard binder (*cf.* gray arrow in Fig. 2), *e.g.*, once initial categories have been identified, and can be achieved via a standard supervised approach. Ultimately, this allows for *dynamically* finetuning NCB's concept representations. Let us now introduce how human stakeholders can inspect and revise NCB's learned concepts.

### 3.3 Inspecting and Revising NCB's Concepts

**Inspection.** NCB inherently enables: (i) *implicit*, (ii) *comparative*, (iii) *interventional* and (iv) *similarity*-based inspection (*cf.* Fig. 3). Where the first three aim at investigating NCB's explicit, *symbolic* concept space (stored in $\mathcal{R}$), the last one aims at investigating its latent, continuous concept space (stored in $\theta$). **(i) Implicit inspection** queries the model to provide a set of examples for a specific concept. Essentially, this answers the question *"What are examples of this concept?"*. NCB answers this question in two ways: by providing samples from the retrieval corpus corresponding to *exemplars* of the concept or by identifying additional data samples belonging to the concept at hand. **(ii) Comparative inspection**, on the other hand, allows comparing two specifically different concepts, *e.g.*, *"Why does this object depict concept H5 and not concept H1?"*. NCB hereby provides examples for both concepts for the user to compare and potentially identify dissimilar properties. Ultimately, this form of inspection allows to answer questions of the form "Why *not* ...?" and represents a valuable tool for in-depth and targeted concept inspection. **(iii) Interventional inspection** allows to answer questions such as *"What if this object would have concept H1?"* To answer this question, NCB utilizes its decoder $g'$. Specifically, by swapping the block $z_i^j$ of a data sample's block-slot encoding with that of a representative sample, $(\mathrm{enc}_l^j, v_l) \in \mathcal{R}^j$, NCB can provide an *interventional* image reconstruction, from which the effect of the swapped concept can be observed. Ultimately, this form of inspection allows to answer important questions of the form "What if ...?". Finally, **(iv) Similarity inspection** allows inspecting NCB's *continuous* encoding space on a more global level (in comparison to the more symbolic, sample-based inspection above), *e.g.*, *"What are similar concepts to this concept?"*. Specifically, NCB's distance metric $d$ directly provides information about the similarity between concepts in the continuous representation. Inspecting the block-slot encoding space thus allows to identify a suboptimal soft binding, *e.g.*, when block encodings are similar according to $g$ but not according to the human stakeholder. Overall, these inspection mechanisms allow a human stakeholder to ask a diverse set of questions concerning a model's learned concepts (*cf.* Fig. 15, Fig. 16 and Fig. 17 for additional examples of the inspection types).

**Revise.** Let us now describe how a human stakeholder can revise NCB's concept space. Below, we provide details on the three main actions for *symbolic* revision (*i.e.*, revision on the representations in $\mathcal{R}$): (i) *merging*, (ii) *deleting*, or (iii) *adding* information. These actions can be performed on a single encoding or on a concept level and essentially represent a form of "reorganization" of information

Table 1: **Comparison of different approaches for concept learning.** Hereby, we differentiate based on the following categories: whether a method (1) is learned in an unsupervised fashion, (2) provides object-level concepts (*i.e.*, can explicitly process multiple objects), (3) provides factor-level concepts (*e.g.*, the color green), (4) provides continuous concept encodings, (5) provides discrete concept encodings, (6) provides inherently inspectable and (7) revisable concept representations.

| Method | Unsupervised | Obj. level | Factor level | Cont. encs | Disc. encs | Inspectable | Revisable |
|---|---|---|---|---|---|---|---|
| CBM [34] | ✗ | ✗ | ✓ | ✗ | ✓ | ✓ | ✓ |
| NeSyCL [68] | ✗ | ✓ | ✓ | ✗ | ✓ | ✓ | ✓ |
| GlanceNets [50] | ✗ | ✗ | ✓ | ✓ | ✓ | ✓ | ✓ |
| VAE [33] | ✓ | ✗ | ✓ | ✓ | ✗ | ✗ | ✗ |
| VQ-VAE [75] | ✓ | ✗ | ✗ | ✓ | ✓ | ✗ | ✗ |
| SA [43] | ✓ | ✓ | ✗ | ✓ | ✗ | (✓) | ✗ |
| SysBinder [65] | ✓ | ✓ | ✓ | ✓ | ✗ | (✓) | ✗ |
| **Neural Concept Binder** | ✓ | ✓ | ✓ | ✓ | ✓ | ✓ | ✓ |

stored in $\mathcal{R}$. Furthermore, we provide details on how to (iv) *revise the continuous latent space*, which essentially requires finetuning of $g$'s parameters. **(i) Merge Concepts:** In the case that $\mathcal{R}$ contains multiple concepts that, according to additional knowledge (*e.g.*, from a human or other model), represent a joint underlying concept (*e.g.*, two concepts for purple in Fig. 3 (right)) it is easy to update the model's internal representations by replacing the concept symbols of one concept with those of the second concept. Specifically, for block $j$ if concept $m$ should be merged with concept $b$ where $m, b \in \{1, \cdots, N_C\}$, then for all corpus tuples, $(\text{enc}_l^j, v_l) \in \mathcal{R}^j$, we replace $v_l$ with $b$ if $v_l = m$. **(ii) Delete Encodings or Concepts:** If $\mathcal{R}^j$ contains an encoding, $\text{enc}_l^j$, for a specific concept, $m$, that does not match the other encodings of that concept (*e.g.*, a misplaced exemplar) this encoding can simply be deleted from the corpus. Accordingly, if an entire concept, $m$, is identified as suboptimal, one can simply delete all corresponding encodings of that concept. *I.e.*, for all corpus tuples, $(\text{enc}_l^j, v_l) \in \mathcal{R}^j$, we remove the tuple if $v_l = m$. **(iii) Add Encodings or Concepts:** If a specific concept is not sufficiently well captured via the existing encodings in $\mathcal{R}^j$, one can simply add a new encoding, $\hat{\text{enc}}_{l+1}^j$, for the concept, $m$, to the corpus. This leads to an additional entry in the corpus, $(\hat{\text{enc}}_{l+1}^j, m)$. Accordingly, it is also possible to add encodings for an entire concept. Hereby, one gathers block encodings of objects that represent that novel concept and adds these to the corpus as $(\hat{\text{enc}}_{l+1}^j, b)$ with $b = N_C + 1$. **(iv) Revise the (Continuous) Latent Space:** Lastly, if the soft binder provides suboptimal object- and factor-level block-slot encodings, it is further possible to integrate revisory feedback on the soft binder's continuous latent space. This can be achieved via additional finetuning of the soft binder's parameters, $\theta$, *e.g.*, via standard forms of weak supervision [42, 69] or interactive learning [68, 61].

In summary, our novel Neural Concept Binder framework fulfills several important desiderata for concept learning (*cf.* Tab. 1). Specifically, NCB learns concepts in an unsupervised fashion that are structured on both an object and factor-level. Furthermore, next to standard continuous encodings, NCB also provides discrete concept representations, which are crucial for interpretability and integration into symbolic computations. Lastly, NCB's concept space is inspectable and revisable, essential for unsupervised learned concept representations.

# 4 Experimental Evaluations

In our evaluations, we investigate the potential of NCB's soft and hard binding mechanisms in unsupervised concept learning and its integration into downstream tasks. Notably, NCB encompasses concept processing between both of its components (soft binder and hard binder) whereby the direction "soft binder ← hard binder" (*cf.* Fig. 2) represents a standard approach (*i.e.*, supervised learning of the soft binder's encoding space via symbolic concept labels, *e.g.*, [34, 68]). Therefore, we focus our evaluations on NCB's more novel processing direction, "soft binder → hard binder". We aim to answer the following research questions: **(Q1)** Does NCB provide **expressive** and **distinct** encodings? **(Q2)** Can NCB be combined with **symbolic** methods to solve complex downstream tasks? **(Q3)** Can NCB's learned concepts be **revised** to improve suboptimal behaviour? **(Q4)** Can NCB be combined with **subsymbolic** methods to *transparently* solve complex downstream tasks?

**Data.** We focus our evaluations on different variations of the popular CLEVR dataset. Specifically, we investigate (Q1 & Q3) in the context of the CLEVR [30] and CLEVR-Easy [65] datasets. For investigating the integration of NCB into symbolic modules (Q2), we utilize our novel CLEVR-Sudoku puzzles introduced in the following. Finally, to evaluate the integration of NCB into subsymbolic modules (Q4), we evaluate on confounded and non-confounded variants of the CLEVR-Hans3 dataset [68]. We provide further details on these datasets in the supplements (*cf.* Suppl. C).

**CLEVR-Sudoku.** To investigate the potential of integrating NCB's discrete concept representations into symbolic downstream tasks, we introduce the novel CLEVR-Sudoku dataset. This dataset presents a challenging visual puzzle that requires both visual object perception and reasoning capabilities. Each sample in the dataset (*cf.* Fig. 4 for an example puzzle) consists of a Sudoku puzzle (partially filled) with CLEVR-based images [30] and additional example images depicting the mapping of relevant object properties to digits. Specifically, each digit in the Sudoku is replaced by an image of an object. All objects representing the same digit share a set of common properties, *e.g.*, in Fig. 4, all objects replacing "1"s are yellow spheres.

We introduce two variants of CLEVR-Sudoku: *Sudoku CLEVR-Easy* and *Sudoku CLEVR*. In the first variant, shape and color are distinguishing properties for the digits. In *Sudoku CLEVR*, additional object attributes — size and material — are relevant for the digit identification. Moreover, up to 10 example images are provided per digit mapping; the fewer examples provided, the more difficult it becomes to learn the mapping. The initial state and digit-attribute mappings vary across samples. One specific intricacy of CLEVR-Sudoku is that the puzzle can only be solved if all subcell images are correctly mapped to their corresponding digits. Even a single mistake can render the Sudoku unsolvable. Thus, compared to standard Sudoku puzzles, which primarily require deductive reasoning, solving CLEVR-Sudoku also demands complex object recognition and the ability to map visual concept perceptions to the *task concepts* (*i.e.*, the 9 digits of Sudoku). For further details, we refer to Suppl. B.

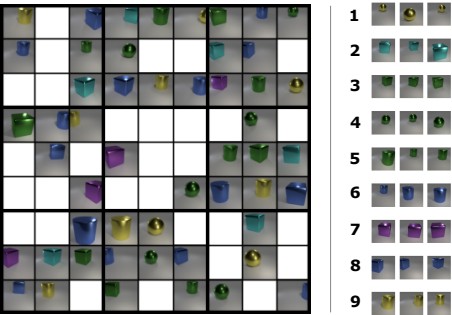

Figure 4: **Example from CLEVR-Sudoku.** Each digit is represented by CLEVR objects with the same attribute combination. The objective is to solve the Sudoku only based on the initial grid of CLEVR images and the digit mapping of candidate examples.

**Models.** For our evaluations, we instantiate Neural Concept Binder based on the SysBinder model [65] for the soft binder encoder, $g$, and HDBSCAN [12, 13] for the clustering model, $h$. Further details about the instantiation can be found in Suppl. A.4. In the context of **Q1**, we compare NCB's results to four variations of the SysBinder model [65], as well as the recent Neural Language of Thought Model (NLOTM) [80]. We refer to the original SysBinder configuration as *SysBinder (cont.)*, which provides continuous block-slot encodings. In *SysBinder*, SysBinder's continuous encodings are discretized at inference time via an `argmin` operation over its internal codebooks. *SysBinder (hard)* is trained from the beginning to produce discrete encodings using a low codebook softmax temperature. *SysBinder (step)* is trained with a step-wise decrease in temperature (*cf.* Suppl. D for details). For evaluations on CLEVR-Sudoku (**Q2** and **Q3**), we first infer NCB's discrete concept-slot encodings from the puzzle's candidate examples. These encodings, along with their corresponding digit labels, are then passed to a symbolic classifier, which is trained to predict digits from the encodings. The classifier subsequently infers the digits for each subcell in the puzzle's initial state. These predictions are used by a constraint propagation and search-based algorithm [55, 10] to solve the puzzle (*cf.* Suppl. E.2 for details). We refer to the combination of the symbolic classifier and constraint solver as the *solver*. We compare the solver's performance when provided with ground-truth (GT) object-property labels (*GT concepts*), encodings from a supervised slot attention encoder [43] (*SA (supervised)*), and the discrete encodings from *SysBinder* (denoted as *SysBinder (unsupervised)*). For classification evaluations (**Q4**), we evaluate a configuration in which a set transformer classifier [37] is provided with NCB's concept encodings (*NCB + NN*) to make final class predictions (*cf.* Suppl. E.4). We compare this to *SA + NN*, where a supervised slot attention encoder [43] provides object-property predictions.

**Metrics.** We evaluate all models based on their accuracies on held-out test splits, each with 3 seeded runs. We provide average accuracies and standard deviations over these. When assessing the expressiveness of NCB's concept-slot encodings (Q1), we evaluate the accuracy for object-property

Table 2: **NCB's concept encodings are expressive despite information bottleneck.** Classifying object properties from different continuous and discrete encodings. The classifier is provided with different amounts of training sample encodings. The best ("•") and runner-up ("○") results are bold.

| Dataset | N Train | SysBinder (cont.) | SysBinder | SysBinder (hard) | SysBinder (step) | NLOTM | Neural Concept Binder |
|---------|---------|-------------------|-----------|------------------|------------------|-------|-----------------------|
| CLEVR-Easy | N=2000 | • **99.83**±0.24 | 92.49±5.45 | 22.92±0.00 | 95.76±4.92 | 84.36±8.54 | ○ **99.02**±1.00 |
| | N=200 | • **99.20**±0.41 | 87.90±8.05 | 22.92±0.00 | 92.42±7.32 | 72.99±8.43 | ○ **98.50**±1.80 |
| | N=50 | ○ **91.13**±4.21 | 78.41±8.69 | 22.92±0.00 | 70.64±11.89 | 49.94±4.97 | • **95.87**±2.93 |
| | N=20 | ○ **64.88**±10.89 | 62.61±7.18 | 22.92±0.00 | 54.61±9.57 | 37.05±4.11 | • **94.22**±4.11 |
| CLEVR | N=2000 | • **98.86**±1.15 | 86.22±10.40 | 36.46±0.00 | 88.90±14.81 | 54.10±18.78 | ○ **97.26**±2.67 |
| | N=200 | • **97.61**±2.58 | 81.13±12.39 | 36.46±0.00 | 83.17±17.05 | 50.17±16.26 | ○ **96.80**±3.01 |
| | N=50 | ○ **93.25**±4.62 | 61.67±8.51 | 36.46±0.00 | 68.81±17.74 | 43.60±13.38 | • **94.67**±4.65 |
| | N=20 | ○ **79.11**±8.75 | 49.79±6.73 | 36.46±0.00 | 58.58±16.09 | 41.52±12.90 | • **88.57**±4.68 |

prediction. When evaluating the performance of the downstream tasks, we provide the percentage of solved CLEVR-Sudokus (Q2) and the classification accuracy on the test set of CLEVR-Hans3 (Q4).

## 4.1 Evaluations

**Discrete, yet expressive representations (Q1).** First, we investigate how much valuable information NCB's discrete concept-slot encodings contain, despite NCB's inherent information bottleneck. To assess this, we train a classifier on NCB's encodings to predict corresponding object-property labels, *e.g.*, the color green (*cf.* Suppl. E.1 for details). In Tab. 2, we present the results for the CLEVR-Easy and CLEVR datasets, using classification training sets with 2000, 200, 50, or 20 encodings. Focusing first on the results for $N = 2000$, we observe that, as expected, the continuous representation of the original SysBinder model contains more information compared to all discrete encodings. Remarkably, however, NCB's discrete concept representations are nearly on par with the continuous encodings. This is particularly notable given NCB's immense information bottleneck[1]. Additionally, we observe that NCB's encodings significantly outperform all other forms of discrete representations. Shifting focus to the results when the classifier is trained on data subsets, we observe a substantial degradation in performance when using encodings from any of the discrete baselines or the continuous encodings. In stark contrast, when classifying based on NCB's encodings, the accuracy remains nearly constant, even with just 1% of the initial training samples. We provide additional ablations on the effect of concept encoding types and NCB's selection function in Suppl. F.1, as well as an ablation analysis on the effect of suboptimal behavior from NCB's individual components in Suppl. F.2. Further analysis of NCB's concept space can be found in Suppl. F.3, along with qualitative examples of learned concepts in Suppl. G. Overall, our results demonstrate the expressiveness of NCB's concept encodings despite their significant information bottleneck. Furthermore, our results suggest that NCB's encodings are easier to generalize compared to the baselines. Thus, we answer **Q1** affirmatively.

**Utilizing unsupervised concepts for solving visual Sudoku (Q2).** In our following evaluations, we investigate the potential of NCB's representations for solving complex reasoning tasks through their integration into symbolic computations. These evaluations are based on our novel CLEVR-Sudoku dataset. The percentage of solved puzzles for CLEVR-Sudoku is reported in Fig. 5. It is important to note that the solver can only solve a puzzle if each image in the initial state has been classified correctly, meaning the results in Fig. 5 represent "all-or-nothing" outcomes. Focusing on the results to the left of the dashed lines, we observe that the symbolic module solves every puzzle with ground-truth (GT) concepts, even when only one example image is provided. Interestingly, performance drops significantly when using encodings from *SA (supervised)*. This highlights the difficulty of the CLEVR-Sudoku puzzles: minor errors in digit prediction can lead to major failures in solving the puzzle. When comparing the performance of encodings from the two unsupervised models, we observe that NCB's concept encodings perform quite well. *E.g.*, they enable solving approximately 50% of the puzzles for the 10-example configurations, compared to approximately 61% for *SA (supervised)*. In contrast, when using *SysBinder*'s encodings, the solver fails across all Sudoku variations. This demonstrates the effectiveness of NCB's binding mechanisms over those of the SysBinder approach alone. We refer to Suppl. F.4 for further discussions and quantitative digit classification results (Fig. 10). Overall, our evaluations highlight the potential of NCB's unsupervised concept encodings for solving complex symbolic downstream tasks. We therefore answer **Q2** affirmatively.

---

[1]*SysBinder (cont.)* provides 2048-sized continuous encodings, whereas NCB provides discrete encodings of size $\leq 16$.

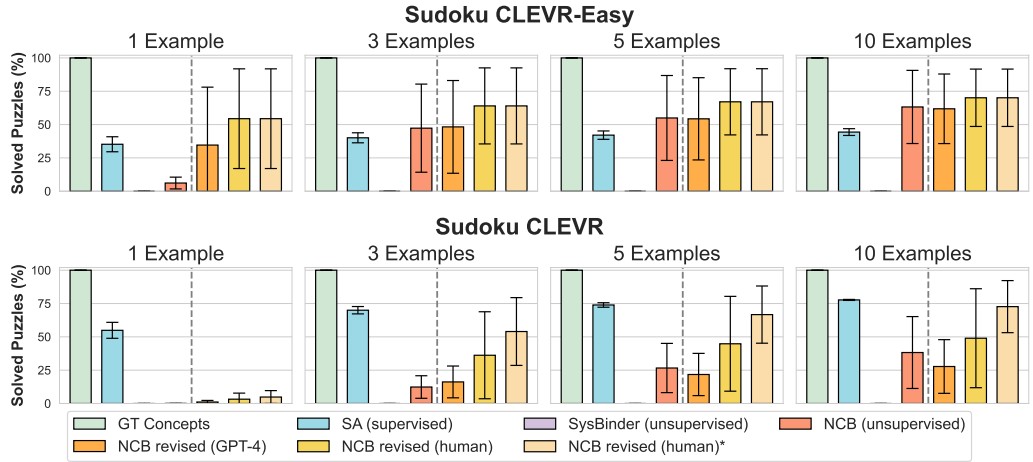

Figure 5: **NCB's unsupervised concepts allow solving symbolic puzzles.** Accuracy of solved Sudokus via different discrete concept encodings on Sudoku CLEVR-Easy and Sudoku CLEVR (left sides). Additional revision on NCB's concepts leads to improved performances (right sides).

Table 3: **NCB's unsupervised concept representations facilitate interpretable neural computations.** Explanations of a NN classifier trained on the unsupervised concepts of NCB. Via NCB's inherent inspection procedures a human stakeholder can identify which concepts the classifier focuses on to make its predictions and thus interpret the NN's underlying decision rule.

| GT Class Rule | NN Expl. | Human Inspection | Human Interpretation |
|---|---|---|---|
| Large, gray cube | C4 ∧ H5 ∧ K5 ∧ O13 ∧ P6 | (Gray1) ∧ (Red ∨ Gray2) ∧ (Large) ∧ (Gray3) ∧ (Gray4) | "A large gray object" |
| Small, metal cube | B4 ∧ D4 ∧ H1 ∧ I1 ∧ K1 | (Cube) ∧ (Small1) ∧ (Small2) ∧ (Small3) ∧ (Small4) | "A small cube" |
| Large, blue sphere | B1 ∧ C7 ∧ H4 ∧ O1 ∧ P2 | (Sphere) ∧ (Blue1) ∧ (Blue2) ∧ (Small ∨ Blue3) ∧ (Blue4 ∨ Green ∨ Purple) | "A blue sphere" |

**Easily revising NCB's concepts (Q3).** In our next evaluations, we illustrate the potential of NCB's revision procedures. Since revising the continuous latent space of NCB's soft binder is analogous to existing approaches (*e.g.*, [61, 58, 69]), we focus on the novel, NCB-specific forms of *symbolic* revision, *i.e.*, revisions within the hard binder's concept space. We demonstrate two forms of symbolic revision (*removing* and *merging* concept information) using feedback from two sources: a pretrained vision-language model (here via GPT-4 [56]) and simulated human feedback. In both cases, we ask the revisory agent to identify which concepts in each block should be removed or merged based on exemplar images of each concept, *i.e.*, implicit concept inspection (*cf.* Suppl. E.3 for details). In Fig. 5, we show CLEVR-Sudoku performance when NCB's retrieval corpus is updated by different revisory agents (*i.e.*, *NCB revised (GPT-4)* and *NCB revised (human)*). Interestingly, while GPT-4's revisions improve performance in settings with few examples, they have a negative impact when more digit examples are present. This is due to GPT-4's suboptimal consistency in object descriptions, leading to the removal or merging of too much concept information. This highlights the potential issue of "ill-informed" feedback (*cf.* Suppl. F.5). In contrast, human revisions provide a substantial boost in Sudoku performance, particularly in puzzle configurations with fewer candidate examples. Moreover, using NCB's similarity inspection mechanism (*cf.* Sec. 3.3), a human stakeholder can easily identify models that suffer from suboptimal soft binding processing. In such cases, these models can be excluded from further downstream evaluations (*cf. NCB revised (human)\**) and refined by finetuning $g$'s parameters (*e.g.*, via approaches from [61, 58, 69]). In Suppl. F.6, we further explore concept revision by *adding* new information. Overall, our results demonstrate the potential and ease of revising NCB's concept space, allowing us to answer **Q3** positively.

**Utilizing unsupervised concepts for understanding neural computations (Q4).** In our final evaluations, we investigate whether NCB's discrete concept encodings can make *subsymbolic* compu-

tations more transparent. We focus on the task of image classification using concept-bottleneck-like approaches [68, 34] on variations of the benchmark CLEVR-Hans3 dataset [68]. While the concept encodings in *NCB + NN* are trained *unsupervised*, they perform on par with the supervised approach of [68] (*cf.* Suppl. F.7). More importantly, integrating NCB's inherently inspectable concept representations into neural computations leads to more transparent decision processes.

We illustrate this in Tab. 3, where we provide class-level explanations of the classifier in *NCB + NN* (*cf.* Suppl. E.4 for details). Using NCB's inspection mechanisms, human stakeholders can easily identify the classifier's internal decision rules for a class (*e.g.*, "a large gray object"). This is a critical feature for deploying trustworthy AI models in real-world scenarios. The key result is that this transparency is achieved even with *unsupervised* concept encodings. In Fig. 6, we further investigate whether a NCB-based neural classifier can be revised to mitigate confounders in the CLEVR-Hans3 dataset (*cf.* Suppl. E.4 and Suppl. F.8 for details). The confounding factor

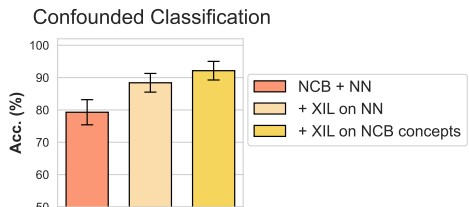

Figure 6: **NCB's unsupervised concept representations facilitate shortcut mitigation.** Test accuracy for classification via NN predictor when trained on *confounded* images.

in the training set is the color *gray*, and we present the non-confounded test set accuracy in Fig. 6. We observe that standard loss-based feedback via explanatory interactive learning (XIL) [68] on the NN classifier's explanations (*+ XIL on NN*) significantly reduces the effect of the confounder. Alternatively, by simply zeroing the activations of the undesired concept *gray* (*+ XIL on concepts*), we achieve even better confounding mitigation results without the typical issues of joint optimization. Our results highlight the potential of integrating NCB's unsupervised concept representations for eliciting transparent and trustworthy subsymbolic computations. We thus answer **Q4** affirmatively.

**Limitations.** NCB largely benefits from high-quality initial block-slot encodings. If these encodings are suboptimal, the resulting concept-slot encodings also degrade in quality. An important next step to handle more complex visual inputs, such as video data, is the integration of recent approaches (*e.g.*, [16, 19]). Additionally, due to NCB's unsupervised training nature, further alignment of NCB's concepts is inevitable for effective deployment in downstream tasks [9]. Further, to build trust in NCB's concept knowledge, human inspection is essential. Lastly, revisions are a critical aspect of NCB. However, they rely on humans to provide accurate feedback; a malicious user could manipulate NCB's concepts. Fortunately, by inspecting the concept space, it is possible to track and mitigate such manipulation effectively.

## 5 Conclusions

In this work, we introduce the Neural Concept Binder framework for learning visual object-factor concepts in an unsupervised manner. Our evaluations suggest that NCB's specific binding mechanisms facilitate the learning of expressive yet discrete concept representations. Furthermore, our results highlight the potential of integrating NCB's inherently inspectable and revisable concept-slot encodings into both symbolic *and* neural modules. Promising directions for future research include exploring the benefits of NCB's representations in continual learning settings [11], high-level concept learning [81], and probabilistic logic programming approaches [66, 67], as well as investigating connections to object-centric causal representation learning [48]. Lastly, incorporating downstream learning signals may be valuable (if present) for improving the quality of NCB's initial concept encodings, *e.g.*, through classification [5, 6] or differentiable clustering [76].

### Acknowledgments

The authors thank Gautam Singh for help with SysBinder and Cyprien Dzialo for preliminary results and insights. This work was supported by the Priority Program (SPP) 2422 in the subproject "Optimization of active surface design of high-speed progressive tools using machine and deep learning algorithms" funded by the German Research Foundation (DFG), the "ML2MT" project from the Volkswagen Stiftung and the "The Adaptive Mind" project from the Hessian Ministry of Science and Arts (HMWK). It has further benefited from the HMWK projects "The Third Wave of Artificial Intelligence - 3AI", and Hessian.AI, as well as the Hessian research priority program LOEWE within the project WhiteBox, and the EU-funded "TANGO" project (EU Horizon 2023, GA No 57100431).

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

# Supplementary Materials

In the following, we provide details on Neural Concept Binder, experimental evaluations as well as additional evaluations.

## Impact Statement

Our work provides a new framework for unsupervised concept learning for visual reasoning. It improves the reliability of the unsupervised concept learning by explicitly including both inspection and revision of the concept space in the framework. NCB thus makes an important step towards more reliable and transparent AI, by providing an interpretable symbolic concept representation. This representation can be utilized within reliable and proven symbolic methods, or to improve transparency of neural modules. However, as the concepts are learned unsupervised, one has to keep in mind that they are not necessarily aligned with human knowledge, and might require inspections to achieve this. As NCB features a concept revision via human feedback, it is also necessary to consider that these revisions could have negative effects. A user with malicious intents could modify the memory and thus make the concept space incorrect. The fact that the learned representation of NCB is explicitly inspectable can, however, prove to be helpful in limiting such malicious interventions.

## A  Details on Neural Concept Binder

### A.1  Details on Systematic Binding and Slot Attention

The binding mechanism (SysBinder) of Singh et al. [65] allows images to be encoded into continuous block-slot representations and relies on the recently introduced slot attention mechanism [43]. In slot attention, so-called slots, $s \in R^{N_S \times N_B D_B}$ (each slot has dimension $N_B D_B$), compete for attending to parts of the input via a softmax-based attention. These slot encodings are iteratively updated and allow to capture distinct objects or image components. The result is an attention matrix $A \in R^{N_S \times D}$ for an input $x \in R^D$. Each entry $A_i$ corresponds to the attention weight of slot $i$ for the input $x$. Based on the attention matrix, the input is processed to read-out each object by multiplying $A$ with the input resulting in a matrix $U \in R^{N_S \times N_B D_B}$.

SysBinder now performs an additional factor binding on the vectors $u_i$ of $U$. The goal of this factor binding mechanism is to find a distribution over a codebook memory for each block in $u_i$, i.e., $u_i^j$. This codebook memory (one for each block), $M^j \in R^{K \times D_B}$, consists of a set of $K$ learnable codebook vectors. Specifically, for each block $j$ an RNN consisting of a GRU and MLP component iteratively updates the $j$-th block of slot $s_i$, $s_i^j$, based on $u_i^j$ and previous $s_i^j$. Finally, a soft information bottleneck is applied where each block $s_i^j$ performs dot-product attention over the codebook memory leading to the final block-slot representation:

$$\mathbf{s}_i^j = \left[ \operatorname*{softmax}_K \left( \frac{\mathbf{s}_i^j \cdot (\mathbf{M}^j)^T}{\sqrt{D_B}} \right) \right] \cdot \mathbf{M}^j$$

This process is iteratively refined together with the refinement processes of slot attention. Overall, the encodings of SysBinder represent each object in an image by a slot with $N_B$ blocks where each block represents a factor of the object like shape or color.

Note that in the main text, the final $s_i^j$ is denoted as $z_i^j$.

### A.2  Selection Function

In the default setting, NCB selects that encoding from the retrieval corpus with the minimal distance to infer a corresponding concept representation. We further explore a top-$k$ approach for the selection function $s$ with $k > 1$. In this case, $s$ selects the values $v_l$, for the $k \in \mathbb{N}$ closest encodings in the retrieval corpus and the resulting $c_i^j$ is obtained via majority vote over these values. Additionally, via this selection approach the probability of $c_i^j$ based on the occurrence distribution over the top-$k$ values $v_l$ can be estimated. We provide ablations regarding this in our evaluations in Suppl. F.1.

**Algorithm 1 Training NCB**: Given a set of images, $X$, a block-slot encoder, $g_\theta$, an unsupervised clustering model $h_\phi$.

| | |
|---|---|
| 1: $\hat{\theta} \leftarrow \texttt{fit}(g_\theta, X)$ | ▷ Step 1: Optimize the block-slot encoder |
| 2: $Z \leftarrow g_{\hat{\theta}}(X)$ | ▷ Step 2.1: Gather block-slots from optimized $g$ |
| 3: $\bar{Z} \leftarrow \texttt{select\_object\_slots}(Z)$ | ▷ Step 2.2: Filter out *non-object* slots |
| 4: **for** $j \in \{1, \cdots, N_B\}$ **do** | |
| 5: $\quad \hat{\phi}^j \leftarrow \texttt{fit}(h, \bar{Z}^j)$ | ▷ Step 2.3: Obtain clustering of $\bar{Z}^j$ |
| 6: $\quad R^j \leftarrow \texttt{distill}(\hat{\phi}^j, \bar{Z}^j)$ | ▷ Step 2.4: Extract clustering representation into $\mathcal{R}^j$ |

## A.3 Details on Training

The first step (*cf.* L.1 in Alg. 1) optimizes the encoder $g$ to provide *object-factorised* block-slot encodings. It is optimized for unsupervised image reconstruction based on the decoder model, $g'_{\theta'} : z \to \tilde{x} \in \mathbb{R}^D$ (*cf.* Fig. 2) and a mean squared error loss: $L = L_{\text{MSE}}(x, g'(g(x)))$. In practice, additional losses have been shown to be beneficial for further improving the obtained block-slot encodings [65, 64].

The goal of NCB's second training step is to obtain the retrieval corpus, $\mathcal{R}$. This procedure is based on obtaining an optimal clustering of block encodings via an unsupervised clustering model $h$ and distilling the resulting information from $h$ into explicit representations in the retrieval corpus. This step is divided into several substeps (*cf.* L.2-6 in Alg. 1). It starts with gathering a set of block-slot encodings $Z = g_{\hat{\theta}}(X)$. As $Z$ can include slots which do not encode objects but, *e.g.*, the background, we first select the "object-slot" encodings from $Z$. This step results in $\bar{Z} \subseteq Z$ and consists of a heuristic selection based on the corresponding slot attention masks (described in the following section).

For each block $j$ we next perform the following steps: (i) a clustering model, $h_{\phi^j}$ (*cf.* Fig. 2), is fit to find a set of clusters within $\bar{Z}^j$ thereby identifying $N_C \in \mathbb{N}$ meaningful clusters. The learning of this optimal clustering is based on an unsupervised criteria, *e.g.*, density based scores [53]. Ideally, this leads to that objects that share similar block encodings are clustered together in the corresponding latent block space, whereas objects that possess very different block encodings are associated with distant clusters. This resulting clustering is stored in $h$'s internal representation which we denote as $\phi^j$ (*e.g.*, the merge tree in a hierarchical clustering method [12, 13, 54]. Importantly, $h_{\phi^j}$ is optimized individually for each block. (ii) In the $\texttt{distill}$ step representative block encodings of each cluster, $\text{enc}^j$, are extracted from $h$'s internal representation, $\phi^j$. Hereby, every $\text{enc}^j$ can represent either an averaged *prototype* or instance-based *exemplar* encoding of a cluster. This is performed for every identified cluster, $v \in \{1, \cdots, N_C\}$ and is based on $\bar{Z}^j$ and $\phi^j$. As a result, the tuples $(\text{enc}^j, v)$ are explicitly stored in the retrieval corpus $\mathcal{R}^j$. The final retrieval corpus consists of the set of individual corpora for each block, $R = [R^1, \cdots, R^{N_B}]$.

We note that in practice, it is further possible to finetune the block-slot encoder, $g$, through supervision from the hard binder, *e.g.*, once initial categories have been identified and can be achieved via a standard supervised approach. Ultimately, this allows for *dynamically* finetuning NCB's concept representations.

**Heuristic object-slot selection.** In the following we describe the process of identifying the slot which contains an object. This is based on heuristically selecting slot ids based on their corresponding slot attention values. Importantly, this approach can select object-slot ids without additional supervision, *e.g.*, via (GT) object segmentation masks.

In principle, our object-slot selection approach finds the slots which contain slot attention values above a predefined threshold, $\delta \in (0, 1]$. However, selecting such a threshold can be cumbersome in practice. In our evaluations we therefore select only a single slot per image, *i.e.*, that slot which contains the maximum slot attention value over all slots. Essentially, this sets the maximum number of selected slots per image to 1 and in images that contain one objects represent no loss of object relevant information. In preliminary evaluations we observed that the consensus between object-slot selection based on GT object segmentation masks (matching object segmentation masks with slot attention masks) and our maximum-based selection heuristic is $99.45\%$ over 2000 single object images.

### A.4 Instantiating Neural Concept Binder

We instantiate NCB's soft binder via the SysBinder approach of Singh et al. [65] which has been shown to provide valuable, object-factor disentangled representations. Thus, the soft binder was trained as in the original setup and with the published hyperparameters. Furthermore we instantiate the clustering model, $h$, via the powerful HDBSCAN method [12, 13, 54] (based on the popular HDBSCAN library[2]). Hereby, $h$'s internal representation, $\phi$, consists of the learned hierarchical merge tree. In practice we found it beneficial to perform a grid search over $h$'s hyperparameters based on the unsupervised density-based cluster validity score [53]. The searched parameters are the minimal cluster size (the minimum number of samples in a group for that group to be considered a cluster) and minimal sample number (the number of samples in a neighborhood for a point to be considered as a core point) each over the values $[5, 10, 15, 20, 25, 30, 50, 80, 100]$. Moreover, we utilize the excess of mass algorithm and allow for single clusters. We performed the training of the retrieval corpus, *i.e.*, fitting $h$, on a dataset of images containing single objects for simplifying the subsequent concept inspection mechanisms of our evaluations. However, this can easily be extended to multiple object images by utilising the soft binder's slot attention masks to identify relevant objects in an image. Finally, we instantiate the retrieval corpus as a set of dictionaries and, unless stated otherwise, we utilise a retrieval corpus which contains one prototype and a set of exemplar encodings per concept. Furthermore, $s_\mathcal{R}$ represents the argmin selection function and we utilize the euclidean distance as $d(\cdot, \cdot)$. It is important to note that $h$ does not make any assumptions about the number of clusters, $N_C$. Thus, although $h$ fits a clustering to best fit the block-slot encodings of a block, it can potentially provide an overparameterized clustering, *e.g.* by representing one underlying factor such as "gray" with several clusters. This highlights the importance of task-alignment, *e.g.*, for symbolic downstream tasks, and concept inspection for general concept alignment. We refer to our code for more details[3], where trained model checkpoints and corresponding parameter logs are available.

### A.5 Computational Resources

The resources used for training NCB were: CPU: AMD EPYC 7742 64- Core Processor, RAM: 2064 GB, GPU: NVIDIA A100-SXM4-40GB GPU with 40 GB of RAM. Hereby, training the SysBinder model [65] is the computational bottleneck of NCB where we utilised two GPUs per SysBinder run. Training for 500 epochs took $\approx$108 GPU hours. The fitting of $h$ (including the grid search over hyperparameters) was performed on the CPU and finished within a few hours.

## B  Details on CLEVR-Sudoku

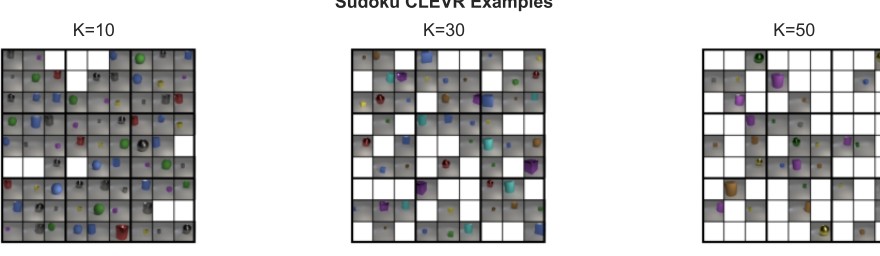

Figure 7: Examples of Sudoku CLEVR for different K values.

CLEVR-Sudoku provides Sudokus based on the datasets CLEVR and CLEVR-Easy. Classic Sudokus have a 9x9 grid which is filled with digits from 1 to 9. In CLEVR-Sudoku these digits are replaced by images of objects. Hereby, a digit corresponds to a specific attribute combination, *e.g.*, "yellow" and "sphere". Consequently, digits of the Sudoku are replaced by images of objects with these attribute combinations. These images each contain one object. To indicate, which attributes correspond to which digit, candidate examples of the digits are provided. The number of these examples is a flexible

---

[2]`https://hdbscan.readthedocs.io/en/latest/index.html`
[3]Code available here.

parameter, in our evaluations we used $N \in \{1, 3, 5, 10\}$. Further, the number of images provided in the Sudoku grid is flexible as well. In our main evaluations we only considered CLEVR-Sudokus with $K = 30$, meaning that 51 of the 81 Sudoku cells are filled and 30 are left to complete. For additional investigation we considered values for $K \in \{10, 50\}$ as well. Examples of those Sudokus for Sudoku CLEVR are shown in Fig. 7. The dataset has a number of 1000 samples for *Sudoku CLEVR-Easy* and *Sudoku CLEVR* respectively for each value of $K$. Each sample has a different puzzle and a distinct set of images, no image is used twice for one puzzle[4].

## C  Datasets

**CLEVR.** Briefly, a CLEVR [30] image contains multiple 3D geometric objects placed in an illuminated background scene. Hereby, the objects can possess one of three forms, one of 8 colors, one of two sizes, one of two materials and a random position within the scene.

**CLEVR-Easy.** CLEVR-Easy [65] images are similar to CLEVR images, except that in CLEVR-Easy the size and material is fixed over all objects, *i.e.*, all objects are large and metallic.

**CLEVR-Hans3.** The CLEVR-Hans3 [68] represents a classification dataset that contains images with CLEVR objects where the image class is determined based on the attribute combination of several objects (*e.g.*, an image belongs to class 1 if it contains a large, gray cube and a large cylinder). Furthermore, we utilize a confounded and non-confounded version of CLEVR-Hans3. In the confounded case (*i.e.*, the original dataset) the train and validation set contains spurious correlations among object attributes (*e.g.*, all large cubes are gray in class 1) that are not present in the test set (*e.g.*, large cubes of class 1 take any color). In our evaluations investigating only neural-based classification we utilize the original validation split as the held-out test split and select a subset from the original training split as validation set. Thus, the non-confounded version corresponds to a standard classification setup in which the data distribution is identical over all three data splits. Lastly we provide evaluations on a single object version of CLEVR-Hans3 (class 1: a large, gray cube; class 2: a small metal cube; class 3: a large, blue sphere; *cf.* Tab. 3) and the original, multi-object version.

## D  Baseline Models

We note upfront, that all SysBind configurations below were trained for as many epochs as NCB, followed by an additional finetuning for 2 epochs on the same dataset that was used to distill NCB's retrieval corpus.

**SysBind (cont.).** This denotes the original SysBinder configuration which was trained as in [65] and provides continuous block-slot encodings. We refer to the original work for hyperparameter details.

**SysBind.** This denotes a SysBinder configuration that was trained as in [65]. However, at inference time we perform discretisation via an argmin operation over the attention values to each block's prototype codebook.

**SysBind (hard).** This denotes a configuration in which the SysBinder model was trained via a codebook attention softmax temperature of $1e - 4$, resulting in a learned discrete representation.

**SysBind (step).** SysBinder (step) is trained by step-wise decreasing this temperature his denotes a configuration in which the SysBinder model was trained via a step-wise decreasing codebook attention softmax temperature (with a decrease by a factor of $0.5$ every 50 epochs, starting from $1.$).

**NLOTM.** NLOTM [80] builds on the principles of SysBinder and incorporates a Semantic Vector-Quantized (SVQ) Variational Autoencoder along with the Autoregressive LoT Prior (ALP). The SVQ component facilitates discrete semantic decomposition of a scene by learning hierarchical, composable factors that correspond closely to objects and their attributes in visual scenes. We refer to the original work for details.

**Supervised Concept Learner.** This corresponds to a slot attention encoder [43] that was trained for set prediction (*i.e.*, in a supervised fashion) to predict the object-properties for every object in a CLEVR image. We refer to Locatello et al. [43] and Stammer et al. [68] for details.

---

[4]The code for generating the dataset is available in our code repository, the already generated data files are accessible under `https://huggingface.co/datasets/AIML-TUDA/CLEVR-Sudoku`

# E  Details on Experimental Setup

## E.1  Classifying object-properties from concept encodings

For our evaluations in the context of (Q1) we utilise a decision tree as classification model that is trained on a set of concept encodings to predict corresponding object properties, *e.g.*, *sphere*, *cube* or *cylinder*. Importantly, we train a separate classifier for each property category, *e.g.*, the categories *shape*, *color*, *material* and *size* in the case of CLEVR, and average accuracies over these. The classifiers parameters correspond to the default parameters of the sklearn library[5].

## E.2  CLEVR-Sudoku evaluations

For our CLEVR-Sudoku evaluations we use a solver that combines a symbolic classifier with a constraint propagation based algorithm. To solve CLEVR-Sudokus, it is at first required to detect the underlying mapping from the object attribute combinations to the digits via the provided candidate examples. For this, we require a symbolic classifier to learn this mapping, which in the case of our evaluations is achieved via a decision tree classifier. For each evaluated model the concept encodings of the candidate example images of a CLEVR-Sudoku are retrieved and provided as input to the classifier. Hereby, the corresponding digits are the labels to be predicted. With the predictions of the trained classifier the concept encodings of the images in the Sudoku grid are classified to get a symbolic representation of the Sudoku, *i.e.*, map the images in the cells to their corresponding digits. Based on this numerical representation of the puzzle, we use an algorithm from [55] that uses a combination of constraint propagation [10] and search. The algorithm keeps track of all possible values for each cell. Within each step, the Sudoku constraints are used to eliminate all invalid digits from the possibilities. Then the search of the algorithm select a digit for a non-filled cell. Based on this digit, the possibilities are updated for all other cells. When there is a constraint violation, the search-tree is traversed backwards and other possible digits for non-filled cells are explored. This process is repeated until the Sudoku is solved (in case the initial state inferred from the objects was correct) or until there is no possible solution left (meaning that the initial state was incorrectly inferred from the objects). The implementation of the algorithm is based on the code from[6]. Finally, to avoid errors due to random seeding of the classifier, for each puzzle we fit 10 independent classifiers (each with different seeds) to predict the corresponding mapping. For the results in our evaluations we average the performance over these 10 classifier seeds.

Lastly, the evaluations in the context of (Q2) are based on the trained (NCB) models of (Q1).

## E.3  Obtaining Revisory Feedback

We note that the evaluations in the context of (Q3) are based on the trained NCB models of (Q1).

**Revisory feedback for downstream Sudoku task.**

To revise its discrete concepts, NCB offers the possibility to delete or merge clusters in the blocks. In the case of merging, the prototypes and exemplars of the clusters to be merged get aggregated so that they all map to the same concept symbol. For deletion there are several processing cases, depending on how many categories are in the block and how many are supposed to be deleted:

- Case 1: if all clusters from a block should be deleted (or if there is only one concept in the block, which should be deleted), we map all samples to the same concept. This results in the block containing no information (we keep the block to avoid issues with the dimensions of the concept representation).

- Case 2: all clusters but one are to be deleted. In this case we still want to distinguish between the presumably "informative" cluster and the uninformative other clusters. Therefore we map all the blocks to be deleted to one cluster id instead of deleting them completely.

- Case 3: at least two clusters should not be deleted. In this case, we completely remove the encodings of the to-be-removed clusters. The cluster id for these clusters no longer exists in the retrieval corpus.

---

[5]`https://scikit-learn.org/stable/modules/generated/sklearn.tree.DecisionTreeClassifier.html`

[6]`https://github.com/ScriptRaccoon/sudoku-solver-python/tree/main`

**Feedback via GPT-4.** We systematically prompt GPT-4 [56] for receiving revisory feedback. We provide example prompts in Listing 1. First, we ask GPT-4 to name relevant object properties for a set of example images, *e.g.*, "shapes: [cube, cylinder], color: [red, blue]". Based on these provided property lists we ask GPT-4 to provide a descriptive list of each exemplar object's image for each concept of each block, *e.g.*, "{Exemplar1: [cube, red], Exemplar2: [cube, blue], ... }". Based on these descriptions we identify whether all exemplar objects of one concept share a common subproperty, *e.g.*, "cube". If there is no common subproperty, the concept should be removed from the retrieval corpus. In a second step we evaluate whether all exemplar objects from two separate concepts share a common subproperty. In this case we decide to merge the concepts based on GPT-4's analysis. We finally integrate GPT-4's feedback into NCB's retrieval corpus via the procedures described above.

**Feedback via simulated humans.** To simulate feedback by a human user, we utilise a decision tree (DT) classifier to classify attributes of objects based on NCB's discrete concepts (similar to Q1). For this, we transform the concept-slot encodings into multi-hot encodings. We then extract the importance of the concepts from the trained DT classifier. Based on this we select "unimportant" concepts to be deleted based on the procedures describe above. Note that in this setting we do not query for feedback considering the merging of concepts.

### E.4  Neural Classification

We note that the evaluations in the context of (Q4) are based on the trained NCB models of (Q1).

**Neural classifier.** In the context of the classification evaluations (Q4) we utilize the setup of Stammer et al. [68]. Specifically, a set transformer [37] is trained to classify images from the CLEVR-Hans3 dataset given encodings that are, in turn, obtained from either NCB or a supervised trained slot attention encoder [43] (SA). In the case of utilizing NCB's encodings we transform the concept-slot encodings into multi-hot encodings to match those of the SA-based setup. We refer to Stammer et al. [68] and our code for additional details concerning this setup.

**Obtaining explanations from the neural classifier.** We provide the explanations in Tab. 3 for the single object version of Fig. 14. To obtain these explanations for the neural classifier we utilize the approach of Stammer et al. [68] which is based on the integrated gradients explanation method [70]. This estimates the importance value of each input element (in this case input concept encodings) for a classifiers final decision. We remove negative importance values and normalise the importance values as in [68]. We then sum over the importance values corresponding to images of a class, normalise the values per block and binarize these aggregated and normalised importance values via the threshold of $0.25$ (*i.e.*, importance values above $0.25$ are set to $1$, otherwise $0$). This provides us with a binary vector indicating which concepts are considered important per block. We illustrate these investigations via explanations from one model.

**Explanatory interactive learning (XIL).** Explanatory interactive learning (+ *XIL on NN*) is used to mitigate the confounder in the CLEVR-Hans dataset. Hereby, (simulated) human feedback on the explanation of the neural classifier is used to retrain the classifier via the loss based approach of Stammer et al. [68]. The feedback annotations mark which of NCB's concepts should *not* be used for the NN's classification decision. This is integrated into the NN by training the model to provide (integrated gradients-based) explanations that do not focus on these concepts. We refer to Stammer et al. [68] for details. The second form of interactive learning (+*XIL on NCB concepts*) is directly applied on the NCB's concept representation. Specifically, concepts from NCB that encode information concerning the irrelevant, confounding factors are simply set to zero, corresponding to *not being inferred for the object in the image*. *E.g.*, if the NCB infers concepts concerning the color "gray" to be present in an object and the underlying confounder is the color "gray" the corresponding concept activations of the NCB's prediction are set to zero, *i.e.*, no gray. Then the neural classifier is retrained on the new concept representations. Next to a better performance, the advantage of this approach is that it does not require the more costly loss-based XIL training loop. We illustrate these investigations via interactions on one model.

Table 4: Ablation of NCB's selection components for classifying attributes from concept representations. Best results are in bold.

| N Train | NCB (P) | NCB (P+E) | NCB (P+E, topk) |
|---------|---------|-----------|-----------------|
| — CLEVR-Easy — | | | |
| N=2000 | $98.76_{\pm1.05}$ | $\mathbf{99.02}_{\pm1.00}$ | $98.93_{\pm1.10}$ |
| N=200 | $97.11_{\pm2.16}$ | $\mathbf{98.50}_{\pm1.80}$ | $98.42_{\pm1.91}$ |
| N=50 | $94.31_{\pm4.47}$ | $\mathbf{95.87}_{\pm2.93}$ | $95.72_{\pm3.04}$ |
| N=20 | $90.50_{\pm7.09}$ | $\mathbf{94.22}_{\pm4.11}$ | $94.15_{\pm4.14}$ |
| — CLEVR — | | | |
| N=2000 | $96.77_{\pm2.63}$ | $\mathbf{97.26}_{\pm2.67}$ | $97.17_{\pm2.68}$ |
| N=200 | $96.41_{\pm2.64}$ | $\mathbf{96.80}_{\pm3.01}$ | $96.80_{\pm3.04}$ |
| N=50 | $94.29_{\pm4.78}$ | $\mathbf{94.67}_{\pm4.65}$ | $94.10_{\pm5.25}$ |
| N=20 | $87.55_{\pm5.35}$ | $\mathbf{88.57}_{\pm4.68}$ | $88.42_{\pm4.63}$ |

Table 5: Ablation: Classifying attributes from concept representations with sub-optimal NCB components. The left column serves as a reference and represents the configurations used in the main evaluations, *i.e.*, where the soft binder was trained for 600 epochs and the clustering model represented the HDBSCAN approach that was optimized via a grid-search over its corresponding hyperparameters.

| N Train | NCB | NCB (50 epochs) | NCB (100 epochs) | NCB (w/o grid search) | NCB (kmeans) |
|---------|-----|-----------------|------------------|-----------------------|--------------|
| — CLEVR — | | | | | |
| N=2000 | $97.26_{\pm2.67}$ | $95.19_{\pm1.2}$ | $94.91_{\pm3.45}$ | $97.69_{\pm2.95}$ | $97.26_{\pm2.80}$ |
| N=200 | $96.80_{\pm3.01}$ | $93.69_{\pm1.08}$ | $93.83_{\pm2.90}$ | $96.80_{\pm3.09}$ | $96.01_{\pm3.51}$ |
| N=50 | $94.67_{\pm4.65}$ | $89.10_{\pm4.29}$ | $89.67_{\pm6.95}$ | $94.46_{\pm5.65}$ | $87.65_{\pm8.78}$ |
| N=20 | $88.57_{\pm4.68}$ | $83.46_{\pm6.08}$ | $88.48_{\pm2.36}$ | $90.51_{\pm4.40}$ | $73.52_{\pm10.92}$ |

# F  Additional Quantitative Results

## F.1  Encoding Expressivity

In our evaluations in Tab. 2 it appears that training for discrete encodings via *SysBinder (hard)* leads to no learning effect of the model altogether. In contrast training step-wise via *SysBinder (step)* provides better results, even slightly above the encodings of *SysBinder* (*i.e.*, training for continuous representations and then discretising via argmin). Lastly, we observe that NCB's encodings lead to much lower performance variance compared to all baselines. Particularly *SysBinder (step)*'s high variances, hint towards issues with local optima.

We further provide ablations in the context of (Q1) on different component choices of NCB in Tab. 4. Specifically, we investigate the effect of a top-$k$ selection function as well as the influence of using only prototype encodings in the retrieval corpus (NCB (P)) versus using prototype *and* exemplar encodings (NCB (P+E)). Unless noted otherwise, the NCB configurations in Tab. 4 utilize the argmin selection function. We note that when using prototypes, the average encoding of all elements in a cluster is formed, resulting in one prototype encoding per cluster in $R^j$. In the second variant, we extend the prototypes with exemplars for each cluster. Exemplars are representative encodings for this cluster added to the corpus, resulting in a larger corpus, which potentially provides an improved structure of the encoding space. Indeed, we observe that NCB provides the best performances via the argmin selection function and utilizing both prototype and exemplar encodings. This was the setting used in all evaluations of the main paper.

## F.2  Ablation Analysis of Suboptimal NCB Components

Lastly, in the context of (Q1) we further refer to ablations in Tab. 5 on the specific implementation choices of the NCB instantiation of our evaluations. We hereby investigate the effect of sub-optimal soft and hard binder components on a classifier's ability to identify object attributes from NCB's concept encodings. Specifically, we investigate (i) the effect when the soft binder, *i.e.*, SysBinder encoder, was trained for fewer epochs, resulting in less disentangled continuous representations, and

(ii) when the HDBSCAN model of the hard binder was not optimized via a parameter grid search or replaced with a more rudimentary clustering model, *i.e.*, a k-means clustering approach [45].

In the leftmost column of Tab. 5, we provide the performances of the NCB configuration of our main evaluations as a reference. As a reminder, hereby, NCB's soft binder was finetuned for 500 epochs, and its hard binder component contains a clustering model based on the HDBSCAN approach that was furthermore optimized via a grid search over its corresponding hyperparameters. Focusing on the next two columns right of the baseline, we observe that when the soft binder component is trained for fewer epochs than the baseline NCB we indeed observe a decrease in classification performance. Notably, however, we still observe higher performances in comparison to the discrete SysBinder configurations (*cf.* Suppl. F.1), but also when compared to SysBinder's continuous configuration (for $N = 20$). Focusing next on the rightmost column of Tab. 5 where NCB's clustering model was replaced with the more rudimentary k-means clustering approach, we observe a strong decrease in classifier performance. This is particularly true in the small data regime ($N = 50$ and $N = 20$). Surprisingly, focusing on the second to the rightmost column, we observe that when we select the default hyperparameter values of the HDBSCAN package (rather than performing a grid-search over these), the classifier reaches slightly improved performances than via the baseline NCB configuration (particularly for $N = 20$). Thus, in this particular case, the default values seem practical. However, this cannot be guaranteed in all future cases, and we still recommend performing a form of grid search if no prior knowledge can be provided upfront on an optimal parameter set. We postulate that the specific density-based cluster validity score used for selecting the optimal cluster parameters has been sub-optimal and leave investigating other, more optimal selection criteria for future work.

Overall, our ablation investigations indeed indicate that we obtain less expressive concept encodings via NCB with less powerful sub-components. However, we also observe a certain amount of robustness of our NCB instantiation towards sub-optimal components.

### F.3 Analysis of Learned Concept Space

We here provide a brief analysis of NCB's learned concept space. These evaluations were performed on the models that were trained in the context of (Q1). Specifically, in Fig. 8, we provide the number of obtained concepts over all blocks (averaged over the 3 initialization seeds) both for CLEVR-Easy and CLEVR. We observe a much larger number of concepts overall for the CLEVR dataset but also a much larger variance in the number of concepts. This is largely due to that in CLEVR-Easy $N_B = 8$ whereas in CLEVR $N_B = 16$. Thus, the models are able to learn a more overparameterized concept space in the case of CLEVR. Further, in Fig. 9, we present the distribution of the number of concepts per block over all 3 NCB runs, both for CLEVR-Easy and for CLEVR. We observe that while most blocks contain maximally 20 concepts for CLEVR-Easy and 50 for CLEVR, there are several block outliers which contain a much greater set of concepts.

These represent cases in which the initial block-slot encoding space was uninformative to begin with and, therefore, difficult to find some form of useful clustering via $h$. Where some of these blocks only contained irrelevant information in general, some blocks encoded positional information, which represents a continuous variable to begin with and is thus unlikely to be well represented via a clustering.

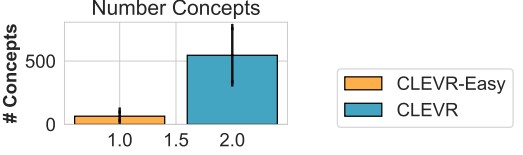

Figure 8: Average number of concepts (over all blocks) in NCB's retreival corpus.

### F.4 CLEVR-Sudoku Evaluations

In our evaluations on (Q2) we observe that, interestingly, for Sudoku CLEVR the supervised object classifier shows better results than for CLEVR-Easy. This seems counter-intuitive, however, in CLEVR-Easy-Sudoku digit labels are mapped to combinations of attributes that only stem from two categories, shape and color (in contrast to four categories in CLEVR-Sudoku) thus making it more likely to obtain recurring attributes over several digits (*e.g.*, digits 3, 4 and 5 of Fig. 4 all depict green objects). Thus, if an error occurs in the digit classification due to errors concerning one attribute the effect of this error will have a larger effect. Moreover in the case of CLEVR-Easy, we observe that in comparison to the supervised model, whose property misprediction errors can lead to large issues in the downstream module, NCB's unsupervised and somewhat overparameterised concept space

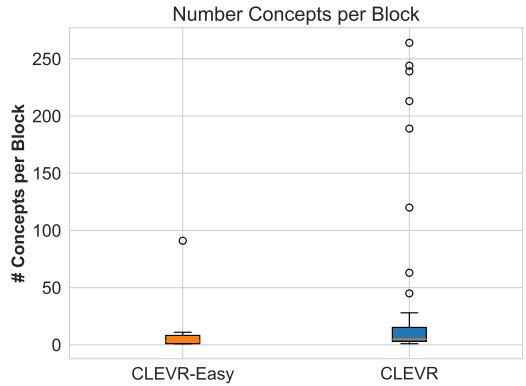

Figure 9: The distribution of number of obtained concepts per block both for CLEVR-Easy and CLEVR. These values are computed over all seeds.

## Sudoku CLEVR-Easy

## Sudoku CLEVR

Figure 10: Error ratios (%) of the digit classification in CLEVR-Sudoku based on different symbolic concept encodings.

appears to dampen this issue, thus leading to a higher number of solved puzzles, *e.g.*, for 3, 5 or 10 examples.

In Fig. 10 we report the errors in predicting the underlying digits of the CLEVR-Sudokus. We observe that the errors of *SysBinder (unsupervised)* are drastically higher than the errors of the other methods. These high classification errors further explain this method's low performances, *i.e.*, did not allow to solve any Sudoku. It can further be seen that for one example per digit the digit classification errors are much higher. This is reasonable as hereby the difficulty for the classifier is also higher. However, with an increasing number of examples the classifier's errors decrease. The relations between the errors in the digit prediction and the overall performance in CLEVR-Sudoku are similar which is sensible since the error is decisive for the number of solved puzzles.

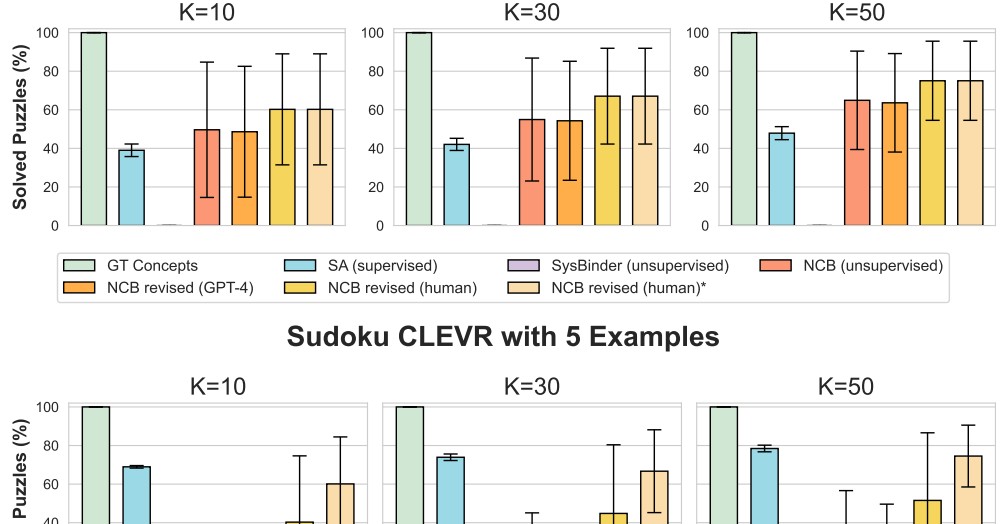

Figure 11: Solved Sudokus (%) of Sudoku CLEVR-Easy and Sudoku CLEVR with different values for K (empty cells).

We further evaluate the influence of the number of missing images per Sudoku. For this we consider Sudokus with $K \in \{10, 30, 50\}$. The results on these variations with 5 candidate example images are reported in Fig. 11. We see that the more empty cells there are in a Sudoku's initial state (higher $K$), the more Sudokus are solved. This is due to the lower probability of misclassifying an image inside the Sudoku cells, as there are less images to classify. This pattern is observable for all of the different concept encodings we compared.

### F.5 Revision Statistics

We provide statistics of the number of resulting removal requests per agent in Fig. 12. For the revision of CLEVR-Easy concepts we can see that GPT-4 detects only a few concepts to delete while via simulated human revision more concepts get deleted. In our initial evaluations (*cf.* Fig. 4) we had observed that human revision leads to

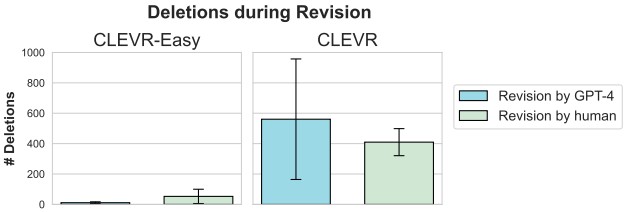

Figure 12: Average number of cluster deletions over all blocks via GPT-4 and simulated human user revision.

substantial improvements while GPT-4's revision even reduces performances slightly. For CLEVR-Sudoku in Fig. 12, we specifically observe that the overall number of deletions via GPT-4 is significantly higher. Interestingly, GPT-4 detects on average more blocks to delete here but also has a higher variance over the 3 different NCB runs. We hypothesize that this very "conservative" revision leads to the removal of concepts that actually contain valuable concept information, thus leading to less expressive concept encodings overall. Ultimately, this is due to mistakes in GPT-4's analysis of provided images (*cf.* Suppl. E.3).

### F.6 Dynamically Discretising Continuous Factors via Symbolic Revision

In our second set of evaluations in the context of (Q3) we investigate the third form of symbolic revision as introduced in Sec. 3.3: adding concept information to the hard binder's retrieval corpus. Hereby, we focus on the task of learning a novel concept that had only been stored implicitly in the soft binder's representations, but not explicitly in the hard binder's representations. Specifically, we focus on positional concepts of CLEVR objects where the underlying GT position is represented via continuous values. Overall, it is debatable whether one, in principle, should or even can represent such a continuous underlying feature via a discrete concept representation.

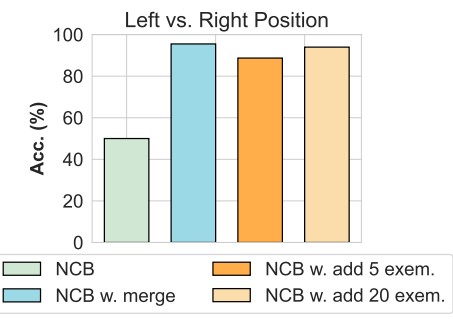

Figure 13: Test accuracy (%) for classifying objects as placed left or right in a scene.

In this set of evaluations we investigate a setting in which it is necessary to identify coarse categorisations of an object's position, *e.g.*, whether the object is placed in the left or right half of an image. We hereby simulate a human stakeholder that, having identified the block $j$ that generally encodes position information, revises the corresponding concept encodings. This revision is performed in two ways: (i) by iterating over all of the block's concepts and merging concepts into left and right concepts or (ii) by replacing all information in $\mathcal{R}$ with encodings from a selected set of positive example images for the two relevant positions. Fig. 13 presents the results of training a classifier to predict the attributes "left" and "right" from NCB's encodings (we here focus only on one seeded run for illustrations) with different types of revision. We observe that both allow to easily retrieve relevant information from NCB's newly revised concept space. These results illustrate the important ability to easily adapt the hard binder's concept representations by *dynamically re-reading* out the information of the soft binder's representations in a use-case based manner. The results further illustrate the effect of adding prior knowledge to NCB's concept representations, thereby potentially reducing the amount of inspection effort required on the stakeholder's side, *e.g.*, in comparison to the merge revision.

### F.7 Classifying CLEVR-Hans3

In our final evaluations (Q1) we highlight the advantage of NCB's concept encodings when combined with *subsymbolic* (*i.e.*, neural) modules for making their decisions transparent.

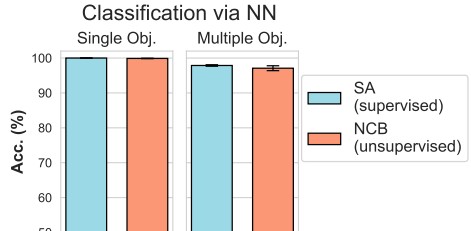

Specifically, while a discrete concept representation is technically not required for neural modules, it has a key advantage: a discrete and inspectable representation allows for transparent downstream computations. We highlight this property in the context of image classification on variations of the benchmark CLEVR-Hans3 dataset [68]. For these evaluations we revert to training a set transformer [37] (denoted as *NN* in the following) for classifying images when provided the unsupervised concept encodings of NCB as image representations. We denote this configuration as *NCB + NN* and compare it to a configuration in which the set transformer is provided concept encodings from a supervised slot attention encoder, denoted as *SA + NN*. In Fig. 14 we obseve that NCB's concepts perform on par with those learned supervisedly, each reaching held-out test accuracies higher than 95%.

Figure 14: Test accuracy (%) for classifying CLEVR-Hans3 images with a neural classifier that is provided concept representations of NCB and of a supervised trained slot attention encoder. We differentiate here between class rules based on one object and multiple objects.

### F.8 Confounding Evaluations

For the confounding mitigation evaluations in the context of (Q4) we train the *NCB + NN* configuration on the confounded version of CLEVR-Hans3, where we hereby focus on the single object class rules similar to those in Fig. 14. In this case all

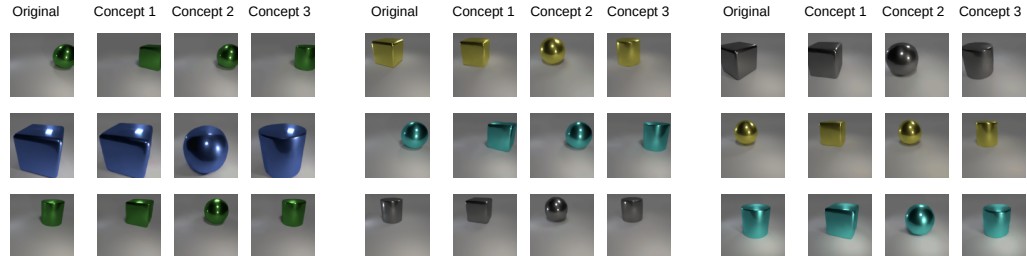

Figure 15: Further examples of **interventional inspection**. By swapping the encoding of block *2* with different exemplar encodings from different concepts, the shape (which is encoded by block *2*) is changed. When swapping the encoding with an exemplar of the same concept, the shape remains unchanged.

large cubes of class one images posses the color gray at training time, but arbitrary colors at test time. We observed accuracies of *NCB + NN* on the confounded validation set of 99.22% against the non-confounded test set 79.29%. This very high validation accuracy versus a significantly reduced test accuracy indicates that the classifier is strongly influenced by the datasets underlying confounding factor.

## G  Qualitative Results

Fig. 15 further exemplifies the inspection types of Sec. 3.3. Fig. 16 and Fig. 17 represent qualitative inspection results of NCB's learned concepts. We specifically present implicit inspection via exemplars of concepts from two blocks from NCB when trained on CLEVR-Easy. One can observe that block 2 (Fig. 16) appears to encode shape concepts, however contains one ambiguous concept. We further observe that Fig. 17 appears to encode color concepts, whereby it contains one ambiguous concept (concept 8) and two concepts that appear to both encode the color purple (concept 9 and 10) which could potentially be merged.

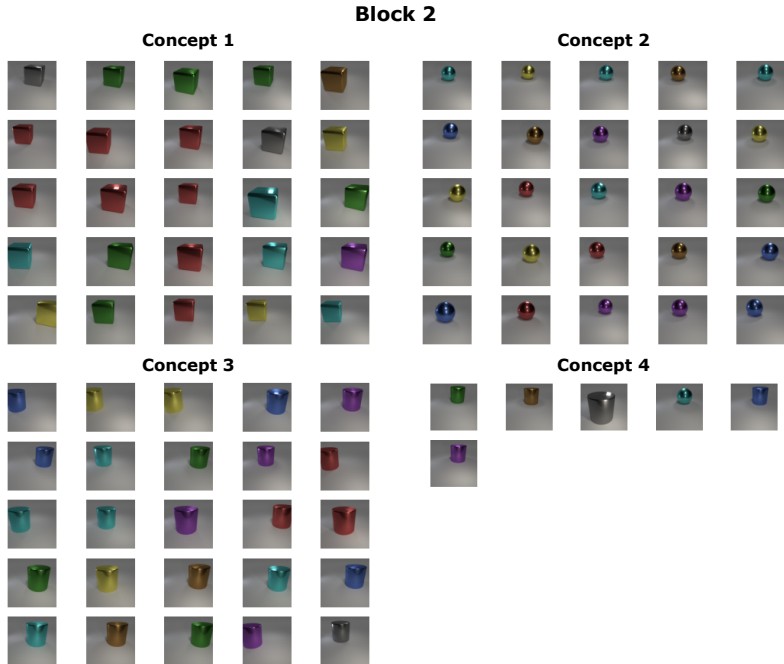

Figure 16: Concepts of Block 2 for NCB with CLEVR-Easy. We here provide **implicit inspection** examples (*i.e.*, via exemplars of each concept). We observe that block 2 appears to encode shape information (concept 1-3) and contains one ambiguous concept (concept 4).

# Block 8

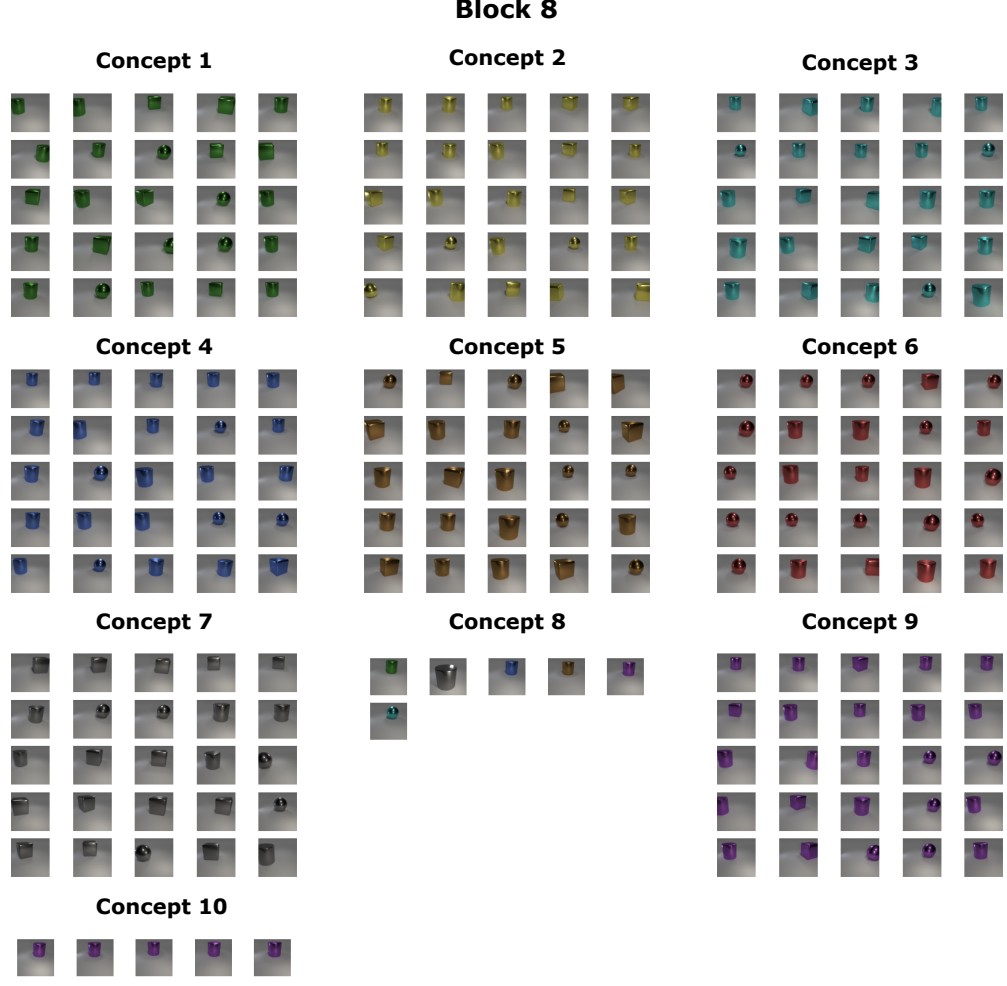

Figure 17: Concepts of Block 8 for NCB with CLEVR-Easy. We here provide **implicit inspection** examples (*i.e.*, via exemplars of each concept). We observe that block 8 appears to be encoding color information, contains one ambiguous concept (concept 8) and two concepts that appear to both encode the color purple (concept 9 and 10).

Table 6: Percentage of solved CLEVR-Sudokus for different number of example images.

| Sudoku CLEVR-Easy | 1 Example | 3 Examples | 5 Examples | 10 Examples |
|---|---|---|---|---|
| GT Concepts | $100.0 \pm 0.00$ | $100.00 \pm 0.00$ | $100.00 \pm 0.0$ | $100.00 \pm 0.00$ |
| SA (supervised) | $35.22 \pm 5.63$ | $40.07 \pm 3.76$ | $42.07 \pm 3.14$ | $44.36 \pm 2.54$ |
| SysBinder (unsupervised) | $0.00 \pm 0.00$ | $0.00 \pm 0.00$ | $0.00 \pm 0.00$ | $0.00 \pm 0.00$ |
| NCB (unsupervised) | $6.13 \pm 4.42$ | $47.30 \pm 33.06$ | $54.95 \pm 31.86$ | $63.21 \pm 27.45$ |
| NCB revised (GPT-4) | $34.61 \pm 43.46$ | $48.25 \pm 34.80$ | $54.31 \pm 30.85$ | $61.83 \pm 26.10$ |
| NCB revised (human) | $54.41 \pm 37.40$ | $64.00 \pm 28.53$ | $67.07 \pm 24.83$ | $70.10 \pm 21.52$ |
| NCB revised (human)* | $50.81 \pm 45.38$ | $62.00 \pm 34.78$ | $66.40 \pm 30.38$ | $70.43 \pm 26.35$ |
| Sudoku CLEVR | | | | |
| GT Concepts | $100.0 \pm 0.00$ | $100.0 \pm 0.0$ | $100.0 \pm 0.00$ | $100.0 \pm 0.00$ |
| SA (supervised) | $54.9 \pm 5.99$ | $69.99 \pm 2.76$ | $73.92 \pm 1.69$ | $77.71 \pm 0.38$ |
| SysBinder (unsupervised) | $0.00 \pm 0.00$ | $0.00 \pm 0.00$ | $0.00 \pm 0.00$ | $0.00 \pm 0.00$ |
| NCB (unsupervised) | $0.01 \pm 0.00$ | $12.36 \pm 8.46$ | $26.62 \pm 18.47$ | $38.24 \pm 26.97$ |
| NCB revised (GPT-4) | $1.11 \pm 1.19$ | $16.18 \pm 11.93$ | $21.76 \pm 15.84$ | $27.75 \pm 20.14$ |
| NCB revised (human) | $3.23 \pm 4.55$ | $36.19 \pm 32.64$ | $44.8 \pm 35.58$ | $48.97 \pm 37.13$ |
| NCB revised (human)* | $4.84 \pm 4.82$ | $54.0 \pm 25.43$ | $66.69 \pm 21.46$ | $72.68 \pm 19.53$ |

Table 7: Error ratios on digit classification of CLEVR-Sudokus for different number of example images.

| Sudoku CLEVR-Easy | 1 Example | 3 Examples | 5 Examples | 10 Examples |
|---|---|---|---|---|
| GT Concepts | $0.00 \pm 0.00$ | $0.00 \pm 0.00$ | $0.0 \pm 0.00$ | $0.0 \pm 0.00$ |
| SA (supervised) | $7.23 \pm 0.89$ | $5.25 \pm 0.16$ | $4.89 \pm 0.09$ | $4.55 \pm 0.09$ |
| SysBinder (unsupervised) | $88.69 \pm 0.05$ | $88.12 \pm 0.18$ | $87.66 \pm 0.31$ | $87.30 \pm 0.40$ |
| NCB (unsupervised) | $23.57 \pm 1.02$ | $3.55 \pm 2.47$ | $2.16 \pm 1.74$ | $1.35 \pm 1.13$ |
| NCB revised (GPT-4) | $13.21 \pm 9.89$ | $3.21 \pm 2.56$ | $2.1 \pm 1.64$ | $1.36 \pm 1.06$ |
| NCB revised (human) | $4.94 \pm 5.68$ | $1.45 \pm 1.49$ | $1.12 \pm 1.05$ | $0.95 \pm 0.83$ |
| NCB revised (human)* | $6.55 \pm 6.37$ | $1.79 \pm 1.73$ | $1.31 \pm 1.24$ | $1.06 \pm 1.00$ |
| Sudoku CLEVR | | | | |
| GT Concepts | $0.00 \pm 0.00$ | $0.00 \pm 0.00$ | $0.00 \pm 0.00$ | $0.00 \pm 0.00$ |
| SA (supervised) | $3.78 \pm 0.79$ | $1.85 \pm 0.18$ | $1.52 \pm 0.09$ | $1.30 \pm 0.08$ |
| SysBinder (unsupervised) | $88.81 \pm 0.03$ | $88.67 \pm 0.03$ | $88.68 \pm 0.1$ | $88.71 \pm 0.11$ |
| NCB (unsupervised) | $48.54 \pm 5.22$ | $11.61 \pm 3.79$ | $6.30 \pm 4.52$ | $4.49 \pm 4.37$ |
| NCB revised (GPT-4) | $34.10 \pm 6.51$ | $9.85 \pm 4.93$ | $8.11 \pm 4.85$ | $7.16 \pm 4.79$ |
| NCB revised (human) | $42.32 \pm 14.56$ | $7.55 \pm 5.84$ | $4.80 \pm 4.94$ | $3.91 \pm 4.38$ |
| NCB revised (human)* | $39.85 \pm 17.31$ | $3.69 \pm 2.57$ | $1.34 \pm 0.97$ | $0.84 \pm 0.66$ |

# H  Numerical Results

In our evaluations we presented the results on CLEVR-Sudoku in the form of bar plots. We refer to Tab. 6, Tab. 7 and Tab. 8 for the numerical values for the different variations of the dataset.

Table 8: Percentage of solved CLEVR-Sudokus for different values of K with 5 example images.

| Sudoku CLEVR-Easy | K=10 | K=30 | K=50 |
|---|---|---|---|
| GT Concepts | $100.0 \pm 0.00$ | $100.0 \pm 0.00$ | $100.0 \pm 0.00$ |
| SA (supervised) | $39.02 \pm 3.25$ | $42.07 \pm 3.14$ | $47.89 \pm 3.37$ |
| SysBinder (unsupervised) | $0.00 \pm 0.00$ | $0.00 \pm 0.00$ | $0.00 \pm 0.00$ |
| NCB (unsupervised) | $49.64 \pm 35.07$ | $54.95 \pm 31.86$ | $64.91 \pm 25.50$ |
| NCB revised (GPT-4) | $48.62 \pm 33.92$ | $54.31 \pm 30.85$ | $63.62 \pm 25.52$ |
| NCB revised (human) | $60.23 \pm 28.77$ | $67.07 \pm 24.83$ | $75.05 \pm 20.51$ |
| NCB revised (human)* | $60.50 \pm 35.24$ | $66.40 \pm 30.38$ | $73.39 \pm 24.96$ |
| Sudoku CLEVR | | | |
| GT Concepts | $100.00 \pm 0.00$ | $100.00 \pm 0.00$ | $100.00 \pm 0.00$ |
| SA (supervised) | $68.96 \pm 0.65$ | $73.92 \pm 1.69$ | $78.46 \pm 1.72$ |
| SysBinder (unsupervised) | $0.00 \pm 0.00$ | $0.00 \pm 0.00$ | $0.00 \pm 0.00$ |
| NCB (unsupervised) | $21.18 \pm 14.85$ | $26.62 \pm 18.47$ | $34.79 \pm 21.84$ |
| NCB revised (GPT-4) | $17.76 \pm 12.86$ | $21.76 \pm 15.84$ | $29.12 \pm 20.49$ |
| NCB revised (human) | $40.30 \pm 34.34$ | $44.80 \pm 35.58$ | $51.55 \pm 35.05$ |
| NCB revised (human)* | $60.10 \pm 24.36$ | $66.69 \pm 21.46$ | $74.54 \pm 16.02$ |

Listing 1: Prompts for GPT-4.

---

```
––––––––––––––––––––––––––––
Property List Prompt:

You are provided six images. An image contains subimages.
Each subimage depicts one object. Each object represents
a reflective geometric solid that is placed in a neutral
gray background scene with a light source. Furthermore,
each object has multiple properties,
e.g., color, shape, size, material.
Each property can be subdivided into several sub−properties,
e.g., brown is a sub−property of the property color.

Please provide a list of obect properties and
subproperties that are depicted in all images. Ignore
the background and the object's luminance and
reflectivity. Use the following answer template:

{
property: [sub−property, sub−property, ...]
property: [sub−property, sub−property, ...]
...
}

––––––––––––––––––––––––––––
Description Prompt:

You are provided an image. The image contains at most 25 subimages.
Each subimage depicts one object. Each object represents a reflective
geometric solid that is placed in a neutral gray background scene with
a light source. Furthermore, each object has multiple properties,
e.g., color. Each property can be subdivided into several sub−properties,
e.g., green is a sub−property of the property color. The possible
properties and sub−properties are the following:

INSERT_PREVIOUSLY_OBTAINED_PROPERTY_LIST

Focusing only on these properties, please perform the following tasks.
First, for every object in the image please list the sub−properties
from the given lists that the object depicts. Only name the sub−properties
that are given. Please use the following format:

{
Object1: [sub−property, ...],
Object2: [sub−property, ...],
...
}

––––––––––––––––––––––––––––
```

