# OpenReview forum: "Neural Concept Binder"
_NeurIPS.cc/2024/Conference — NeurIPS 2024 poster_

### Official Review · Reviewer_6FRw · 2024-07-10

**Soundness:** 3
**Presentation:** 1
**Contribution:** 3
**Rating:** 5
**Confidence:** 4

**Summary:**

The paper proposes a novel approach to unsupervised concept learning based on both continuous and discrete encodings. Neural Concept Binder (NCB) allows humans inspecting and revising the learnt concepts. In the experiments, NCB’s discrete concept encodings result as expressive as the continuous encodings. Also, NCB can be integrated with symbolic and sub symbolic module. Finally, to support the experimental evaluation the paper introduces a novel dataset CLEVER-Sudoku, very suitable for neuro-symbolic benchmarking.

**Strengths:**

-	**Novelty**: the proposed approach, although based on existing works SySBinder and Slot attention, is surely novel in the field of concept learning and potentially very relevant as it may strongly facilitate the extraction and discovery of unsupervised concepts. Particularly the possibility to revise concepts is completely novel to the best of my knowledge and very useful to improve human-computer interaction.
-	**Novel resource** presented: CLEVER-Sudoku will be surely an important resource for the Neuro-symbolic literature.

**Weaknesses:**

## Major issues:
  * Method presentation:
    - The way in which block-slot-encodings are obtained, is badly presented. Although it is based on previous literature, since it is a key architectural component, it should have been presented more in details. I suggest the author to employ a background section to report the way in which slot attention and sys binder work, in order to make the paper self-contained.
    - Figure 2 which illustrates the core of the method is quite confusing: it is not clear how the discrete concepts are actually represented (the concept slot encodings reported are positive continuos representations). Also, the reminders to the figures in the text do not help as they generically refer to the entire figure and not to a specific block. A color coding of the different parts of the model could help understanding.
    - How does the $\texttt{enc}_l^j$ work is not clear. What does it receive in input? Where it is extracted from?
    - All the revision operations are definitely not clear. The formal operation to be executed is often confusing.
  * Experimental evaluation:
    - Models: NCB has been compared only against SysBinder. While it is a very novel and innovative method, there is a complete lack of benchmarking against standard unsupervised concept-based approaches such as SENN[1], BotCL[2], ACE[3]. Comparing against supervised approaches such as CBM[4] or CEM[5] could have been also useful.
    - Datasets: NCB is only tested on variants of CLEVER. While it is surely an interesting benchmark, real-world benchmarks are missing. Experiments on CUB or CELEBA, for instance, would have been very appreciated to better understand the scalability of the approach.

## Minor issues
  * Related work:
    - The unsupervised concept learning literature does not review several important concept-based paper working both post-hoc and explainable by design. Some examples are SENN[1], ACE[3], VAEL[6], as well as notorious prototype-based approaches such as Prototype layer [7] and ProtopNets[8].
    -  Unlike you state, continuous and discrete representations have been combined in recent literature for supervised concept learning. Some examples are CEM[5] and ProbCBM[9].
  * Unclear sentences:
    - “Briefly, given an image x, NCB derives a symbolic representation, c, which expresses the concepts of the objects in the image, i.e., object-factor level concepts. Herefore, NCB infers a block-slot encoding, z, of the image and performs a retrieval-based discretization step to finally infer concept-slot encodings, c”. The consequentiality of the inference process is misleading from this sentence.
  * Method Inspection. What the authors refer as implicit, comparative, interventional and similarity-based inspections are normally referred to as example-based explanations (implicit and similarity-based) and counterfactual explanations (comparative and interventional). Sticking to well-known terms in literature is a good choice to avoid further confusion in the reader.

Overall, I think it's an interesting paper proposing a novel approach to unsupervised concept learning. However, I think it will benefit from a further revision to deeply improve method presentation and expand the experimental campaing including other standard unsupervised concept-learning approaches and datasets.

[1] Alvarez Melis, David, and Tommi Jaakkola. "Towards robust interpretability with self-explaining neural networks." Advances in neural information processing systems 31 (2018).

[2] Wang, B., Li, L., Nakashima, Y., and Nagahara, H. “Learning bottleneck concepts in image classification”. In Proceedings of the IEEE/CVF Conference on Computer Vision and Pattern Recognition (2023).

[3] Ghorbani, Amirata, et al. "Towards automatic concept-based explanations." Advances in neural information processing systems 32 (2019).

[4] Koh, Pang Wei, et al. "Concept bottleneck models." International conference on machine learning. PMLR, 2020.

[5] Espinosa Zarlenga, Mateo, et al. "Concept embedding models: Beyond the accuracy-explainability trade-off." Advances in Neural Information Processing Systems 35 (2022): 21400-21413.

[6] Misino, Eleonora, Giuseppe Marra, and Emanuele Sansone. "Vael: Bridging variational autoencoders and probabilistic logic programming." Advances in Neural Information Processing Systems 35 (2022): 4667-4679.

[7] Li, Oscar, et al. "Deep learning for case-based reasoning through prototypes: A neural network that explains its predictions." Proceedings of the AAAI Conference on Artificial Intelligence. Vol. 32. No. 1. 2018.

[8] Chen, Chaofan, et al. "This looks like that: deep learning for interpretable image recognition." Advances in neural information processing systems 32 (2019).

[9] Kim, Eunji, et al. "Probabilistic Concept Bottleneck Models." International Conference on Machine Learning. PMLR, 2023.

**Questions:**

A few questions to try to understand better the notation employed to define the revision operations.
- What do the authors mean with $v_l \rightarrow v_m$?
- What does the add operation work and how one can provide an encoding for a concept and be sure the network employs it as intended?

**Limitations:**

The method limitations are well addressed by the authors.

---

> ### Author Rebuttal · Authors · 2024-08-07
>
> **W1** (More background):
>
> We agree that adding more information on Sysbinder and Slot Attention can help the reader and make the paper overall more self-contained. We have added an additional section with details for the camera-ready version and provide it in a comment below.
>
>
> **W2** (Clarity of Figure 2):
>
> We agree and have modified the Figure: (i) we have updated the digits (that had previously represented the concept symbols) in the "Hard Binder", "Retrieval Corpus" and "Concept-slot encodings" components to the notation used throughout the experiments. Specifically, we now denote concepts in Figure 2 with a capital letter for the block and a natural number for the category ID, e.g., A3 for the third concept in the first block (A). (ii) Furthermore, we have added Roman numerals (I-VI) to each component in the figure and reference these specifically within the main text. (iii) We have updated the text accordingly for more clarity. Overall, we agree this greatly helps in understanding the individual components. Apologies for having missed that in the initial version.
>
> **W3** ($\texttt{enc}_l^j$):
>
> Let us go through this question step by step. We base this on the more detailed training descriptions in A.2 (appendix).
> First, NCB infers the block-slot encodings of a set of input images. These are denoted as $\bar{Z}$ in Alg. 1 in the appendix. NCB then performs clustering of these encodings per block, i.e., it clusters the blocks j of each encoding (denoted $\bar{Z}^j$) and assigns each of them one cluster label, $v \in \{1, \cdots, N_C\}$. Per cluster, $v$, NCB next identifies exemplar and prototype encodings (i.e., representative encodings identified via the clustering method and cluster averages).
> Each of these exemplar and prototype encodings correspond to one $\texttt{enc}^j$ in the tuples, $(\texttt{enc}^j, v)$, that are stored in the retrieval corpus $\mathcal{R}^j$ of one block $j$. We now use the index $l$ to identify specific encodings out of $\mathcal{R}^j$, leading to $\mathcal{R}^j := \{(\texttt{enc}_l^j, v_l) : l \in \{1, ..., |R^j| \} \}$. In conclusion, $\texttt{enc}_l^j$ represents one block encoding that is stored in $\mathcal{R}^j$ and has been assigned to a specific cluster $v_l$.
> We agree that highlighing these additional steps benefit the understanding of the main text and have updated it accordingly. We apologize for the brevity in the initial submission.
>
>
> **W4** (Revision operators):
>
> Indeed, we agree that the revision operators are somewhat confusing. It seems the attempt to formalize these steps has overcomplicated things. We have now removed the formal operations from section 3.3 and described these in words. E.g. we have posted the updated $\texttt{merge}$ description in a comment below.
>
> **W5** (Additional baselines):
>
> We fully agree on the value of additional baseline models. We have, therefore, added the recent NLOTM model as a novel baseline for our evaluations. We refer here to our general response and hope the reviewer can agree on the relevance of NLOTM as a baseline model.
>
> However, we politely disagree that the methods suggested by the reviewer represent valuable baselines. Specifically, [1] cannot handle object-level concept learning, nor are the concept assignments discrete (an image is represented as a continuous mixture of concepts). [2] on the other hand, focuses on image regions as concepts. Similar issues hold for [3], i.e., they consider concepts to represent segments of an image and do not provide object-level concepts. Moreover, [3] focuses on post-hoc learning of concepts, which makes potential model revisions tricky and the inspection of the model ambiguous. Overall, we fully agree that the references to unsupervised concept learning are related to our work and we have noted them accordingly in our paper. However, none of them tackle the same problem setting as NCB aims to, i.e., learning discrete, object-level concept representations without supervision. We therefore consider the benefit of running these particular baselines as minor.
>
> Concerning supervised models ([4,5]): in fact, we had compared against supervised concept-based models in the context of Q2 and Q4 (denoted as SA (supervised), c.f. also F.7) whereby we observe that indeed NCBs unsupervised concepts are competitive in comparison to the GT supervised concepts of a slot attention-based CBM despite any supervision in training NCB.
>
>
> **W6** (Additional datasets):
>
> We kindly refer here to the general response above.
>
>
> **W7** (Related work):
>
> Agreed, we have added the missing references. We note that we had already referenced ProbCBM, though not in the context of discrete and continuous. We have now added this to this respective section.
>
> **W8** (Unclear sentence):
>
> We apologize for the confusion here. We have updated the sentence (see comment below).
>
> **W9** (Inspection terminology):
>
> We agree that connecting the terms example-based and counterfactual explanations can help to understand the outcome of the different inspection forms of NCB. We have now denoted comparative and interventional inspection as two ways to obtain forms of counterfactual explanations and implicit and similarity-based inspection as two ways to obtain forms of example-based explanations. Thanks for the hint!
>
> **Q1** ($v_l$→$v_m$):
>
> We have removed this notation. It had previously described replacing $v_l$ with $v_m$. We refer here to the response above concerning revision operators.
>
> **Q2** (add operator):
>
> We have updated the add revision as in a comment below.
> Overall, this operation can be used when the hard binder has missed concepts of the original encoding space. However, if NCB's soft binder has not learned to encode a specific concept at all (i.e., does not represent it in its block encodings) or the concept that a user wishes to add is not present in the dataset at all, one has to revert to the fourth type of revision, i.e., an additional finetuning of the soft binder is necessary.

---

> ### Author Response · Authors · 2024-08-07
> **Additional background section**
>
> **Background**
>
> The binding mechanism (SysBinder) of Singh et al. (2024) allows images to be encoded into continuous block-slot representations and relies on the recently introduced slot attention mechanism [2]. In slot attention, so-called slots, $s \in R^{N_S \times N_B D_B}$ (each slot has dimension $N_B D_B$), compete for attending to parts of the input via a softmax-based attention. These slot encodings are iteratively updated and allow to capture distinct objects or image components. The result is an attention matrix $A \in R^{N_S \times D}$ for an input $x \in R^{D}$. Each entry $A_{i}$ corresponds to the attention weight of slot $i$ for the input $x$. Based on the attention matrix, the input is processed to read-out each object by multiplying $A$ with the input resulting in a matrix $U \in R^{N_S \times N_B D_B}$.
>
> SysBinder now performs an additional factor binding on the vectors $u_i$ of $U$.
> The goal of this factor binding mechanism is to find a distribution over a codebook memory for each block in $u_i$, i.e., $u_{i}^j$. This codebook memory (one for each block), $M^j \in R^{K \times D_B}$, consists of a set of $K$ learnable codebook vectors.
> Specifically, for each block $j$ an RNN consisting of a GRU and MLP component iteratively updates the $j$-th block of slot $s_i$, $s_{i}^j$, based on $u_i^j$ and previous $s_{i}^{j}$. Finally, a soft information bottleneck is applied where each block $s_i^j$ performs dot-product attention over the codebook memory leading to the final block-slot representation:
>
> $$
> \mathbf{s}_{i}^j=\left[\underset{K}{\operatorname{softmax}}\left(\frac{\mathbf{s}_i^j \cdot (\mathbf{M}^j)^T}{\sqrt{D_B}}\right)\right] \cdot \mathbf{M}^j
> $$
>
> This process is iteratively refined together with the refinement processes of slot attention. Overall, the encodings of SysBinder represent each object in an image by a slot with $N_B$ blocks where each block represents a factor of the object like shape or color.
>
> Note that in the main text, the final $s_i^j$ is denoted as $z_i^j$.
>
> ---
> [1] Singh, Gautam, Sungjin Ahn, and Yeongbin Kim. "Neural Systematic Binder." ICLR, 2023.
>
> [2] Locatello, Francesco, et al. "Object-centric learning with slot attention." NeurIPS, 2020.

---

> ### Author Response · Authors · 2024-08-07
> **Updated text on merge revision**
>
> (i) Merge Concepts: In the case that $\mathcal{R}$ contains multiple concepts that represent a joint underlying concept (e.g., two concepts for purple in Fig.3 (right)) it is easy to update the model's internal representations by replacing the concept symbols of one concept with those of a second concept. Specifically, according to human or additional model feedback, if concept $m$ in block $j$ should be merged with concept $b$ ($m,b \in \{1, \cdots, N_C\}$) then for all corpus tuples, $(\texttt{enc}_l^j, v_l) \in R^j$, we replace $v_l$ with $b$ if $v_l = m$.

---

> ### Author Response · Authors · 2024-08-07
> **Updated text on add revision**
>
> (iii) Add Encodings or Concepts: If a specific concept is not sufficiently well captured via the existing encodings in $\mathcal{R}^j$, one can simply add a new encoding for the concept, $m$, to the corpus: $\hat{\texttt{enc}}_{l+1}^j$
>
> This leads to an additional entry in the corpus: $(\hat{\texttt{enc}}_{l+1}^j, m)$
>
> Accordingly, it is also possible to add encodings for an entire concept. Hereby, via the soft binder one infers block encodings of example objects that represent that novel concept, $b$, and adds these to the corpus as $(\hat{\texttt{enc}}_{l+1}^j, b)$ with $b = N_C+1$.

---

> ### Author Response · Authors · 2024-08-07
> **Updated unclear sentence**
>
> Briefly, given an image, $x$, NCB infers latent block-slot encodings, $z$, and performs a retrieval-based discretization step on $z$ to infer concept-slot encodings, $c$. These express the concepts of the objects in the image, i.e., object-factor level concepts.

---

> ### Comment · Reviewer_6FRw · 2024-08-12
>
> I thank the authors for their efforts in trying to address the suggested issues.
>
> However, I will increase my score to 5 only since I still think is a borderline paper:
>
> i) The method presentation should have been improved according to what the authors have reported, but I should review it again to assess it better.
>
> ii) Testing only on toy datasets is still not acceptable for a neurips paper. Although I agree that better object centric learning will improve the performance of the model, how the current method behave on natural image dataset is of crucial importance to globally assess the model. Bad, but promising results would have still been appreciated.

---

> > ### Author Response · Authors · 2024-08-13
> >
> > We thank the reviewer for their time and for reconsidering their rating. However, we disagree that non-synthetic datasets are mandatory for NeurIPS. In fact, several influential and recent NeurIPS papers in the field of object-centric/NeSy learning were published based only on synthetic datasets, e.g., [1,2,3,4].
> >
> > [1] Locatello, Francesco, et al. "Object-centric learning with slot attention." NeurIPS, 2020.
> >
> > [2] van Krieken, Emile, et al. "A-nesi: A scalable approximate method for probabilistic neurosymbolic inference." NeurIPS, 2023.
> >
> > [3] Marconato, Emanuele, et al. "Not all neuro-symbolic concepts are created equal: Analysis and mitigation of reasoning shortcuts." NeurIPS, 2023.
> >
> > [4] Li, Zenan, et al. "Neuro-symbolic learning yielding logical constraints." NeurIPS, 2023.

---

### Official Review · Reviewer_U7bj · 2024-07-11

**Soundness:** 3
**Presentation:** 3
**Contribution:** 1
**Rating:** 4
**Confidence:** 5

**Summary:**

This paper introduces neural concept binder, a neural symbolic framework that utilizes both soft and hard binding. Building on top of the sysbinder model, it can additionally do exemplar-based hard binding and revise concepts. Evaluations made on CLEVR and the proposed CLEVR-Sudoku dataset proved the method's validity.

**Strengths:**

- The paper is well-written and easy to read. Connections to previous works are clarified nicely.

- It's good to see the incorporations of both hard and soft bindings to existing neural-symbolic frameworks.

- The model achieved good performance on the proposed CLEVR-Sudoku task and can do satisfactory concept revision and inspection, which is a neat proof of concept that hard binding works.

**Weaknesses:**

There are several weaknesses I can foresee that may lead to the rejection of this paper.

- Limited contribution: After so many years of developing neural-symbolic methods in visual reasoning, from the earliest modular approaches to unsupervised concept learners, code-based reasoning models, and recent visual programming-like frameworks, the goal of neural-symbolic modeling has dramatically changed. In this work, the neural concept binder still focuses on one of the earliest task categories designed for visual reasoning (CLEVR attribute classifications or unsupervised concept learners). It's also built on top of sysbinder, in other words, it's merely an incremental improvement by adding a retrieval-based library.

- I don't see any generalizability of this method beyond extremely toy tasks (attribute classification). The proposed CLEVR-Sudoku is strange and does not correspond to any of the real-world visual reasoning tasks. Relational tasks are also not tackled in this paper.

**Questions:**

- How generalizable can this method be? Can it serve as a part of the closed-loop reasoning in CLEVR (I mean, the original CLEVR questions, as tackled in NS-CL and NS-VQA line of work)?

- Can relational concepts be similarly represented via soft/hard binding?

**Limitations:**

Yes, the authors have adequately addressed the limitations. I do not see any potential negative societal impact.

---

> ### Author Rebuttal · Authors · 2024-08-07
>
> **W1.1** (NeSy contribution):
>
> We agree that the field of neuro-symbolic AI is rapidly evolving with different focuses, e.g., visual reasoning in real-world images. However, works that are focussing on higher-level neuro-symbolic problems still heavily rely on mapping raw input images to symbolic representations, whether that is by using pre-trained concept extractors [1, 2, 3] or foundation models [4, 5, 6] or forms of weaker supervision [7].
>
> However, how to obtain meaningful symbols from images in a fully unsupervised way remains a core challenge for the community. With our work, we address exactly this open problem, and to our knowledge, only [8] has additionally attempted to tackle this task. We hereby specifically focus on the importance of being able to inspect and effectively revise learned symbols, particularly due to the unsupervised nature of the learning setup. These aspects are most often overlooked in the majority of research that is in the pursuit of performance-driven models. Overall, we believe that our work offers a unique contribution to the field of concept learning, as also all other reviewers agree upon. If the reviewer is aware of specific works that are unknown to us yet relevant for this topic, we would be grateful if they could share them so we can better assess the reviewer's remarks.
>
> **W1.2** (utilizing SysBinder):
>
> We strongly disagree that our work represents only an incremental improvement over SysBinder.
> While in this work, we have focused on utilizing the SysBinder encoder as NCB's soft binder, our method is not limited to it. Instead, this encoder can be replaced by any current or future models that process images into object-factor encodings (e.g., also models that can process natural images).
> With our framework, we propose a general method for discovering discrete symbols from continuous representations that offers inspection and revision possibilities for human stakeholders. This is not trivial and a very important aspect for unsupervised learning that, to our knowledge, has not been addressed by other works.
>
> **W2.1** (attribute classification as "toy task"):
>
> We disagree and consider extracting symbols from (unlabeled) images as a cornerstone for a variety of neuro-symbolic methods. Many neuro-symbolic approaches rely on extracting symbolic representations from the input, whereas most recent works specifically rely on pre-trained models [1, 2, 3, 9]. NCB, on the other hand, allows to learn such symbols **without** supervision and a main goal of our work is to introduce NCB as a core module for other frameworks (as particularly reviewer VMaX has acknowledged), e.g., for more complex visual reasoning methods. With the evaluations in the context of Q2-Q4, we have highlighted this ability.
>
> **W2.2** (CLEVR Sudoku):
>
> With CLEVR Sudoku, we are proposing a very relevant challenge that is combining complex visual inputs with explicit reasoning on a symbolic level.
> Specifically, the underlying reasoning skills required to solve CLEVR Sudoku are very much relevant for real-world scenarios.
> In fact, many works agree that reasoning with abstract representations (i.e., concepts) are essential for robust generalization [10, 11, 12]. Overall, there is a great deal of interest, and thus, a variety of benchmarks that investigate exactly this form of required intelligence [13, 14, 15, 16, 17]. These specifically exclude language priors and common knowledge priors of real-world images to asses a model's learning and reasoning abilities.
>
> **W2.3** (Relational tasks):
>
> The goal of our NCB framework is to extract meaningful symbolic representations from the input, which can be beneficial for a variety of tasks. Hereby, NCB focuses on learning discrete object-level concept representations from images, i.e., focusing on unary relational concepts. The aim overall is not to propose a framework to solve relational tasks. However, NCB can be used as an easily integrable module for models that perform relational reasoning, e.g., which usually require pre-trained, (semi-)supervised concept extractors.
> Overall, we agree that it is important for future work to integrate NCB into more complex relational tasks, as we had noted in our conclusion.
>
> **Q1** (closed-loop reasoning):
>
> Yes, exactly, NCB can be easily integrated into closed-loop reasoning. This is exactly the intended application of our method.
> E.g., NS-VQA had relied on the extraction of the scene representation via a supervised trained scene parser with ground truth object attributes. Instead of this scene parser, one can employ NCB, which discovers attributes without supervision and use the object’s attention masks (from the slot attention component) to determine their position.
> In NS-CL, the concepts are learned via the question-answering task. This is an alternative approach to concept discovery that, however, requires question-answer pairs for the image domain that target those concepts.
>
> **Q2** (relational concepts):
>
> In principle, we do think it is possible to represent relational concepts via soft/hard binding, though we do not investigate this here. As mentioned above, our approach is intended to be integrated into other methods, such as visual programming or program synthesis, where the learned relations would be based on NCB’s concepts. Hereby, some current work focuses learning relations from images neurally, and others focus on symbolic approaches. Thus, whether to perform relational concept learning via neural, soft binding principles or symbolic, hard binding principles, or a combination of these is still up for investigation. We further refer to our response above concerning relational tasks.

---

> ### Author Response · Authors · 2024-08-07
> **References**
>
> [1] Yi et al. "Neural-symbolic vqa: Disentangling reasoning from vision and language understanding." NeurIPS, 2018.
>
> [2] Koh et al. "Concept bottleneck models." ICML, 2020.
>
> [3] Shindo et al. "α ILP: thinking visual scenes as differentiable logic programs." Machine Learning, 2023.
>
> [4] Surís et al. "Vipergpt: Visual inference via python execution for reasoning." CVPR, 2023.
>
> [5] Gupta et al. "Visual programming: Compositional visual reasoning without training." CVPR, 2023.
>
> [6] Hsu et al. "What’s left? concept grounding with logic-enhanced foundation models." NeurIPS, 2024.
>
> [7] Mao et al. "The neuro-symbolic concept learner: Interpreting scenes, words, and sentences from natural supervision." ICLR, 2019.
>
> [8] Wu et al. "Neural Language of Thought Models." ICLR, 2024.
>
> [9] Wüst et al. "Pix2Code: Learning to Compose Neural Visual Concepts as Programs." UAI, 2024.
>
> [10] Mitchell, Melanie. "Abstraction and analogy‐making in artificial intelligence." Annals of the New York Academy of Sciences, 2021.
>
> [11] Zhang et al. "How Far Are We from Intelligent Visual Deductive Reasoning?." ICLR 2024 Workshop: How Far Are We From AGI.
>
> [12] Lake et al. "The Omniglot challenge: a 3-year progress report." Current Opinion in Behavioral Sciences, 2019.
>
> [13] Chollet, François. "On the measure of intelligence." arXiv, 2019.
>
> [14] Moskvichev et al. "The conceptarc benchmark: Evaluating understanding and generalization in the arc domain." arXiv, 2023.
>
> [15] Raven, Jean. "Raven progressive matrices." Handbook of nonverbal assessment, 2003.
>
> [16] Lake et al. "Human-level concept learning through probabilistic program induction." Science, 2015.
>
> [17] Nie et al. "Bongard-logo: A new benchmark for human-level concept learning and reasoning." NeurIPS, 2020.

---

> > ### Comment · Reviewer_U7bj · 2024-08-13
> >
> > Thanks for the authors' response. After reading the rebuttal, I've decided to increase the score by 1.

---

> > > ### Author Response · Authors · 2024-08-13
> > >
> > > We thank the reviewer for the time! However, we kindly ask the reviewer to raise remaining concerns in case we could not resolve all issues with our initial response.

---

### Official Review · Reviewer_VMaX · 2024-07-11

**Soundness:** 3
**Presentation:** 4
**Contribution:** 4
**Rating:** 7
**Confidence:** 3

**Summary:**

The authors introduced a pioneering framework that combines an object-centric learning module with a retrieval-based module to address visual reasoning tasks and a new visual reasoning task, CLEVR Sudoku. The proposed method demonstrated significant potential in effectively acquiring inspectable and revisable concepts via human or machine feedback in various scenarios.

**Strengths:**

- S1: The proposed method offers significant novelty in that it has the potential to serve as a building block for concept learning, which can be leveraged as a core module in other frameworks. The author's well-structured experiments provided compelling evidence in support of these claims.

**Weaknesses:**

- W1: The proposed approach can be interpreted as directly integrating SysBinder and HDBSCAN. Because the initial concept detection fundamentally depends on the complete functionality of SysBinder, this framework may not circumvent specific inherent challenges of object-centric learning, including inductive bias resulting from choosing the proper object-factor encoder and identifiability issues.
- W2: Using HDBSCAN is intuitive in the proposed method, but it would be beneficial to include an additional experiment that compares different clustering methods.

**Questions:**

Please check out the Weakness section first. I listed the following questions and suggestions that would be helpful for authors' future works:
- Q1: The recent method [1] in object-centric learning literature is linked to causal representation learning and the identifiability of slot representations. How can this be integrated into your framework?
- Q2: Object-factor learning can be interpreted as learning atoms in logic, and the NN explanations in Table 2 can be seen as the simple form of propositional logic in a neuro-symbolic framework. How can an object-centric learning framework be extended to represent logical rules, such as in the form of first-order logic?

Reference
- [1] Mansouri, Amin, et al. "Object-centric architectures enable efficient causal representation learning." arXiv preprint arXiv:2310.19054 (2023).

**Limitations:**

Please check out the Weakness and Question sections.

---

> ### Author Rebuttal · Authors · 2024-08-07
>
> **W1** (dependence on continuous encoder):
>
> Indeed, the quality of the initial continuous concept encodings is important for the resulting discrete concept representation. We had remarked on this in the context of our ablations. We have now added an ablation study to highlight this empirically (c.f. Tab.1 (middle) in additional pdf). Specifically, via earlier checkpoints of the sysbinder encoder we observe that the weaker the encoder is, the less expressive are NCB's final discrete concept representations. However, as better models are being developed (e.g., that are shown to also process natural images), one can easily integrate these into the NCB framework and replace the SysBinder encoder that we used for our evaluations.
>
> **W2** (ablation clustering):
>
> Thanks for the hint! We have added an ablation evaluation on the CLEVR dataset in the context of Q1 where we compare the expressiveness of concept representations when utilizing k-means rather than HDBSCAN (see corresponding table in additional pdf and general response above). We observe that for generalization, the HDBSCAN approach performs significantly better.
>
> **Q1** (causal representation learning):
>
> Thanks for the pointer! This work is indeed interesting, as it connects the more “informal” object-centric detections with the guarantees (and also assumptions) of causal representation learning. While this work and NCB could be considered somewhat parallel, some possibilities exist to combine both. One could use the weakly supervised perturbations to finetune NCB's initial soft binder encoder if one wants to move toward weak supervision. On the other hand, one could also utilize the unsupervised trained NCB with its interventional inspection to generate sparse perturbation of images. This could potentially be possible without specific human supervision but might be limited to the concepts NCB discovers unsupervised. Overall, there are many possibilities for future research in this direction. We have added the reference accordingly into the main text.
>
> **Q2** (OC-learning and logic):
>
> That's a very important question for the future development of object-centric and neuro-symbolic learning in general. In fact, one motivation for our proposed framework is to be able to extract unsupervised discrete representations from object-centric representations that can be integrated into symbolic AI approaches such as different logic systems. Specifically, NCB's concepts can already be utilized, to some extent, for first-order logic as properties about different objects are encoded into symbols, i.e., unary relations. Thus, first-order logic formulas containing quantors can be expressed out of the box. However, the NCB framework is currently not aimed at encoding arbitrary n-ary relations. We consider utilizing NCB in approaches like [1] could mitigate this issue to learn more complex logic programs based on the unsupervised learned concepts of NCB.
>
> ---
> [1] Wüst et al. "Pix2Code: Learning to Compose Neural Visual Concepts as Programs." UAI, 2024.

---

> > ### Comment · Reviewer_VMaX · 2024-08-12
> > **Reponse to the author's rebuttal**
> >
> > Thank you for the authors' efforts in providing additional experiments on clustering and the reference.
> > I'm maintaining the score.

---

> > > ### Author Response · Authors · 2024-08-13
> > >
> > > We thank the reviewer for their valuable time and appreciation for our work!

---

### Official Review · Reviewer_edmj · 2024-07-14

**Soundness:** 3
**Presentation:** 2
**Contribution:** 2
**Rating:** 6
**Confidence:** 3

**Summary:**

This paper introduces Neural Concept Binder, a framework for obtaining discrete concept representations from images without any supervision. The method is an extension of Neural Systematic Binder (SysBinder), adding a clustering step on top of the block-slot representations to obtain discrete concept representations. The resulting representations are interpretable and modifiable, as shown in the experiments. The model is additionally evaluated on property prediction and downstream tasks on modifications of the CLEVR dataset and shown to be able to leverage fewer training samples than SysBinder.

**Strengths:**

The paper investigates an important problem in learning discrete, interpretable concepts from images in an unsupervised way. The model is a logical extension of SysBinder in clustering the representations to obtain discrete concepts. The experiments show improvements in sample efficiency of these discrete representations over SysBinder’s continuous representations.

**Weaknesses:**

1. Since the solver used in the Sudoku experiments is the same across all baselines, it seems the determining factor of performance is in how well the digits are classified. Therefore, I do not believe framing this evaluation in the context of Sudoku adds any insight—in fact it seems to add unnecessary noise to the evaluation. The evaluation in the appendix (Figure 8) seems more informative and sufficient for determining the benefit of NCB.
2. This paper is missing several related citations: Unsupervised Concept Discovery Mitigates Spurious Correlations (https://arxiv.org/abs/2402.13368) and Neural Language of Thought Models (https://arxiv.org/abs/2402.01203). NLoTM is particularly relevant and can be an additional baseline since it also extends SysBinder to learn discrete representations, except it is trained in an end-to-end way.
3. The discussion in section 3.3 is interesting, but it would be informative to tie each point with corresponding experimental evidence.

**Questions:**

1. For SysBinder (hard) and SysBinder (step), do the models train well with this temperature adjustment? E.g. do they exhibit decent reconstruction and slot decomposition?
2. I’m not sure if I completely understand the analysis for Q4, but since this is done on the concept encodings, and not the discrete representations, can the same analysis be done with SysBinder representations? If so, does NCB offer any additional benefits here?
3. How important is the choice of clustering algorithm to the results? What if we use a simple k-means clustering as is done in the original SysBinder paper?

**Limitations:**

Yes

---

> ### Author Rebuttal · Authors · 2024-08-07
>
> **W1** (regarding RQ2 (CLEVR Sudoku)):
>
> A determining factor for the performance on CLEVR-Sudoku is the classification of the digits. We agree that this has, to some degree, already been investigated in the context of Q1, where we tested the suitability of NCB’s concept representations for few-shot GT attribute classification. However, in Q2 we wanted to extend these findings into a much more challenging downstream task (than just GT attribute prediction) and specifically focus on the integration into purely symbolic systems that require correct, task-specific symbolic mappings. Importantly, as NCB is trained unsupervised, there is no guarantee that the learned concept symbols will be directly translatable to the symbols required for a downstream task. It is, therefore, potentially necessary to learn a translation from NCB's concept symbols to the task symbols, e.g., individual object attributes to specific object compositions. Overall, this is a very valuable problem setting concerning real-world applications, e.g., in planning (e.g., navigating through a world of objects). In Q2, we investigate NCB's potential for integration in such settings and consider the evaluations on CLEVR-Sudoku to be a helpful illustrative example. Additionally, the evaluations in Q2 also serve as baselines for investigating the ability to revise NCB's concepts for such a complex downstream task in the context of Q3.
>
> Nonetheless, we do acknowledge that Figure 8 also provides valuable insights that fall a bit short in the main paper (thanks for the hint!). We have updated the main text to put more focus on these results within the discussion of Q2.
>
> **W2** (additional citations and baseline):
>
> Thank you for pointing out the related works, we were not aware of them. We consider Arefin et al. orthogonal to our work as they focus on learning object-level concepts in comparison to our object-factor level concepts and have included it in our related works section.
> Indeed, Wu et al. is relevant to our problem setting. Unfortunately, it was published so close to the NeurIPS deadline that we seem to have missed it; thank you for pointing it out! We have provided the results of this model in our general response and included it as an additional baseline in our paper. We agree this helps highlight our work.
>
> **W3** (Evidence for discussion 3.3):
>
> Thank you for raising this point! We agree that the paper would definitely benefit from more illustrations of the results that can be obtained by inspection. We therefore added example visualizations (c.f. additional pdf). Further, in the context of Q3, we do investigate revising models. Specifically, we apply $\texttt{merge}$ and $\texttt{delete}$ revision; the corresponding models are denoted as "NCB revised (GPT-4)" and "NCB revised (human)". We also investigate the application of $\texttt{add}$ in the "left-right" experiment in the appendix (F4).
>
>
> **Q1** (Training SysBinder (hard) and (step)):
>
> Good question! The SysBinder (hard) model did not train well, i.e., the reconstruction was terrible. The SysBinder (step) model, on the other hand, had stable training and provided reconstructions of the quality of the vanilla SysBinder.
> These results support recent findings [2] that training from the start with hard discretization results in badly performing models. I.e., it is very difficult for the model to learn good representations at all (e.g., for reconstruction) with such a strict bottleneck.
> On the other hand, learning step-wise can more easily lead to issues with local optima than for models without any bottleneck.
>
>
> **Q2** (RQ4 evaluations):
>
> We are sorry for the confusion here. Within Q4, we investigate the popular setting of concept-bottleneck-like approaches [1,2] in which a model predicts (discrete) high-level concepts from an image and a second model makes the task prediction based on these concept activations. This leads to models with more transparent and high-level explainability properties. In principle, one can also utilize continuous concept encodings rather than fully discrete concept representations. However, this makes their understandability more ambiguous. In Q4, we investigate the case in which the prediction model is given discrete concept representations. We compare to supervised discrete concept representations (c.f. F7) and particularly investigate the ability to inspect and revise such models that utilize NCB's concepts. Thus, the used concept encodings for the evaluation are actually the discrete concept-slot encodings of NCB. Overall, interpreting and revising models that utilize continuous embeddings is not as straightforward, we have therefore omitted such comparisons here. We have further clarified this now in the text of Q4.
>
> **Q3** (importance of hdbscan):
>
> Thanks for the valid questions! We refer here to the general response above with novel ablation evaluations in the additional pdf.
>
> ---
> [1] Koh et al. "Concept bottleneck models." ICML, 2020.
>
> [2] Stammer et al. "Right for the right concept: Revising neuro-symbolic concepts by interacting with their explanations." CVPR, 2021.

---

> > ### Comment · Reviewer_edmj · 2024-08-11
> > **Response to Rebuttal**
> >
> > Thank you for answering my questions and taking the time to run additional experiments. Overall, I think this is a nice extension of SysBinder, although I am still not convinced that the Sudoku experiments are necessary to demonstrate the model's capabilities. I have decided to increase the score.

---

> > > ### Author Response · Authors · 2024-08-13
> > >
> > > We thank the reviewer for their time and reconsidering their rating!

---

### Author Rebuttal · Authors · 2024-08-07

We thank all the reviewers for their time and valuable feedback. We are especially happy to receive so much positive feedback concerning the importance of the tackled problem ("**investigates an important problem**" - edmj), the contribution of our work overall ("**pioneering framework**" - VMaX, "**novel in the field of concept learning**" - 6FRw) and the novel CLEVR Sudoku dataset ("**important resource for the Neuro-symbolic literature**" - 6FRw). We are also happy to hear our writing is of high quality ("**well-written and easy to read**" - U7bj).

With the Neural Concept Binder, we introduce a framework to **learn discrete, interpretable concepts from images** in an **unsupervised** way. We specifically focus on **object-factor-level** concepts and the ability to **inspect and revise** these concepts. This is achieved by combining an object-centric learning module with a retrieval-based module, where the retriever component is beneficial both to obtain discrete concepts and effectively integrate revisory feedback from a human or second model. In our evaluations, we highlight the expressiveness of NCB's concepts and their potential to be integrated into various model settings (both symbolic and neural downstream modules). Moreover, we introduce CLEVR-Sudoku, a challenging, neuro-symbolic benchmark dataset.

### Additional baselines:
Thanks for the suggestions! We have indeed run novel baseline evaluations in the context of Q1 based on the recent SOTA model NLOTM [5]. To our knowledge, this is the only approach that also allows us to learn discrete, object-centric concept representations from unlabeled images. Two seeds are still pending (due to long training durations). We provide preliminary results for one seed in Tab. 2 (additional pdf) and will provide the missing seeds for the final version. We observe a strong decrease in the ability to infer GT object attributes from NLOTM's discrete representations over those of NCB. This suggests that despite NLOTM's explicit training approach for discrete representations, NCB's approach has its advantage regarding representation expressiveness. However, future investigations are necessary to make conclusive remarks. Overall, these results highlight the difficulty of the task and the ability of NCB to tackle it.

Additionally, our work focuses on the ability to inspect and revise learned concepts. NLOTM, on the other hand, emphasizes the ability to generate novel images. It would be interesting to investigate how both approaches can be combined, e.g., to aid NCB's inspection mechanisms via NLOTM's generation abilities.


### Clustering ablations:
We agree and have added an ablation to investigate the effect of using k-means rather than the more powerful HDBSCAN (c.f. Tab.1 (right) in additional pdf). We observe that, particularly in the small-data regime, the concept representations obtained via k-means are less expressive (though still better than the discrete SysBinder baselines). Overall, the HDBSCAN approach performs better in terms of generalization. However, in principle, the NCB framework is implementation-agnostic to the choice of clustering method, e.g., one can also integrate more powerful recent neural approaches for clustering [1].

### Real-world datasets:
While we fully agree that natural-image datasets are important, we note that the scalability and ability to handle these via NCB depends on the object-centric encoder's (NCB's soft binder) ability to handle natural images. Importantly, this is a general challenge for object-centric learning research and a big topic in the current research community [2,3]. While this line of research is extremely important, it is also somewhat parallel to our work. As object-centric encoders improve, concept discovery via NCB based on these encoders will also be able to handle more natural images. For our evaluations, we mainly focused on instantiating NCB's soft binder with the SysBinder encoder (currently the SOTA model for obtaining object-factor representations without supervision). To the best of our knowledge, SysBinder has only been evaluated on CLEVR-based datasets. Our work focuses on how to extract discrete concept representations from such an encoder (irrespective of the kind of images it can process) and, importantly, how to inspect and revise these representations. Therefore, we have not investigated the scalability of SysBinder to more natural images and consider it out of the scope of this work.

Concerning the scalability of the hard binder component, we expect that learning the hard binder should not lead to particular scaling issues (depending on the number of block encodings to be clustered and the scope of the grid search). In addition, NCB's retrieval can be sped up, e.g., via FIASS [4].

### Additional figures for inspection types:
We have added qualitative images in the appendix (e.g., Fig.1 in add. pdf) to further exemplify the inspection types of section 3.3.


---
[1] Vardakas et al. "Neural clustering based on implicit maximum likelihood." Neural Computing and Applications, 2023.

[2] Singh et al. "Guided Latent Slot Diffusion for Object-Centric Learning." arXiv, 2024.

[3] Elsayed et al. "Savi++: Towards end-to-end object-centric learning from real-world videos." NeurIPS, 2022.

[4] Johnson et al. "Billion-scale similarity search with GPUs." IEEE Transactions on Big Data, 2019.

[5] Wu et al. "Neural Language of Thought Models." ICLR, 2024.

---

### Author Response · Authors · 2024-08-11

Dear Reviewers,

As the discussion period is soon approaching,
we wish to thank everyone for the feedback provided on our manuscript. In light of the revisions and clarifications made, we believe we have addressed the concerns that were previously raised.

Particularly, we would like to highlight the new comparison to the NLOTM baseline (which we have now completed for all seeds; see below) and several ablations on the clustering component of NCB.

We are encouraged by the positive feedback and hope that our responses have resolved the reviewer's questions. In this context, we respectfully ask the reviewers to reevaluate their initial score and are happy to answer any remaining questions or clarifications.

Thank you once again for your time, expertise, and consideration.

### Remaining results for NLOTM
By now, the experiments on the NLOTM baseline are finished for all seeds.
As the preliminary results on the first seed indicated, it is substantially harder to infer the ground-truth concepts from NLOTM's representation than NCB's.

| N Train | NCB | seed 0       |
|---------|---------|--------------|
| N=2000  | 99.02+-1.00 | 84.36+-8.54  |
| N=200   | 98.50+-1.80 | 72.99+-8.43  |
| N=50    | 95.87+-2.93 | 49.94+-4.97  |
| N=20    | 94.22+-4.11 | 37.05+-4.11  |

---

### Decision · Program_Chairs · 2024-09-25

**Decision:**

Accept (poster)

**Comment:**

This work proposes Neural Concept Binder (NCB), a novel framework for unsupervised concept learning. Specifically, the goal is to learn expressive yet inspectable and revisable concepts from unlabeled data. Further, the work introduces new dataset CLEVR-Sudoku dataset to show the utility of the proposed approach.

Summary Of Reasons To Publish:

1) Proposed approach has the potential to serve as a building block for concept learning.

2) Solid experiment results.

3) Introduces challenging dataset representing challenging visual puzzle requiring both perception and reasoning capabilities.


Summary Of Suggested Revisions:

I spent quality time going over reviewer’s concerns and the authors rebuttal. Reviewers pointed out concerns on missing experiment evidence, ablations, and presentation, clarity issues. Authors addressed these major concerns to some extent in the rebuttal.